# Development of an atmospheric chemistry model coupled to the PALM model system 6.0: Implementation and first applications

Basit Khan[1], Sabine Banzhaf[2], Edward C. Chan[2,3], Renate Forkel[1], Farah Kanani-Sühring[4,7], Klaus Ketelsen[5], Mona Kurppa[6], Björn Maronga[4,10], Matthias Mauder[1], Siegfried Raasch[4], Emmanuele Russo[2,8,9], Martijn Schaap[2], and Matthias Sühring[4]

[1]Institute of Meteorology and Climate Research, Atmospheric Environmental Research (IMK-IFU), Karlsruhe Institute of Technology, Garmisch-Partenkirchen, 82467, Germany
[2]Freie Universität Berlin (FUB), Institute of Meteorology, TrUmF, Germany
[3]Institute for Advanced Sustainability Studies (IASS), Potsdam, Germany
[4]Leibniz University Hannover (LUH), Institute of Meteorology and Climatology, Germany
[5]Independent Software Consultant, Hannover, Germany
[6]University of Helsinki, Finland
[7]Harz Energie GmbH & Co. KG, Goslar, Germany
[8]Climate and Environmental Physics, Physics Institute, University of Bern, Sidlerstrasse 5, 3012, Bern, Switzerland
[9]Oeschger Centre for Climate Change Research, University of Bern, Hochschulstrasse 4, 3012, Bern, Switzerland
[10]University of Bergen, Geophysical Institute, Bergen, Norway

**Correspondence:** Basit Khan (basit.khan@kit.edu)

**Abstract.**

In this article we describe the implementation of an online-coupled gas-phase chemistry model in the turbulence resolving PALM model system 6.0 (formerly an abbreviation for Parallelized Large-eddy Simulation Model and now an independent name). The new chemistry model is implemented in the PALM model as part of the PALM-4U (PALM for urban applications) components, which are designed for application of PALM model in the urban environment (Maronga et al., 2020). The latest version of the Kinetic PreProcessor (KPP, 2.2.3), has been utilized for the numerical integration of gas-phase chemical reactions. A number of tropospheric gas-phase chemistry mechanisms of different complexity have been implemented ranging from the photostationary state (PHSTAT) to mechanisms with a strongly simplified VOC chemistry (e.g. the SMOG mechanism from KPP) and the Carbon bond mechanism (CBM4, Gery et al. (1989)), which includes a more comprehensive, but still simplified VOC chemistry. Further mechanisms can also be easily added by the user. In this work, we provide a detailed description of the chemistry model, its structure and input requirements along with its various features and limitations. A case study is presented to demonstrate the application of the new chemistry model in the urban environment. The computation domain of the case study comprises of part of Berlin, Germany. Emissions are considered using street-type dependent emission factors from traffic sources. Three chemical mechanisms of varying complexity and one no-reaction (passive) case have been applied and results are compared with observations from two permanent air quality stations in Berlin that fall within the computation domain. Even though the feedback of the model's aerosol concentrations on meteorology is not yet considered in the current version of the model, the results show the importance of online photochemistry and dispersion of air pollutants in the urban boundary layer for high spatial and temporal resolutions. The simulated $NO_x$ and $O_3$ species show reasonable

agreement with observations. The agreement is better during midday and poorest during the evening transition hours and at night. The CBM4 and SMOG mechanisms show better agreement with observations than the steady state PHSTAT mechanism.

**Keywords.** Urban air quality model; mircoscale chemistry model; Large-eddy simulation chemistry model; gas-phase photo-chemistry; chemical equilibrium; chemical mechanisms.

## 5   1   Introduction

More than half of the world's population lives in cities and the number is expected to exceed two-thirds by the year 2050 (United Nations, 2014). The high population density in urban areas leads to intense resource utilization, increased energy consumption and high traffic volumes which results in large amounts of air pollutant emissions. Various urban features such as heterogeneity of building distribution, large amount of impervious material, scarcity of vegetation, and street geometry can influence the
atmospheric flow, its turbulence regime, and the micro-climate within the urban boundary layer that accordingly modify the transport, chemical transformation, and removal of air pollutants. (Hidalgo et al., 2008). Air pollution has a multitude of complex effects on human health, material, ecology and environment. In order to develop policies and strategies to protect human health and environment, a better understanding of the interaction between air pollutants and the complex flow within the urban built areas is necessary.

Air quality of a given region is strongly dependent on the meteorological conditions and pollutant emissions (Seaman, 2000; Jacob and Winner, 2009). In urban canopies, turbulence can modify pollutant concentrations both within and downstream of urban areas. Interactions between meteorology and chemistry are complex and mostly non-linear. Numerical models are useful tools to capture these interactions and help to understand the effect of meteorology on the chemical processes. Modelling of air quality on the regional scale has made major advances within the past decades (Baklanov et al., 2014). However, small scale
dispersion of pollutants from traffic and other sources within urban areas and their chemical and physical transformation is still poorly understood and difficult to predict due to uncertainty in emissions and complexity of modelling turbulence within and above the urban canopies. Besides, the computational costs for including air pollution chemistry and physics into the models are remarkably high due to additional prognostic equations for chemical species and the corresponding chemical reactions.

Reynolds Averaged Navier–Stokes (RANS) based dispersion models are now widely used for assessing urban air quality by
providing predictions of present and future air pollution levels as well as temporal and spatial variations (Vardoulakis et al., 2003; Sharma et al., 2017). In these models, atmospheric turbulence at the city level is primarily resolved by the Reynolds-Averaged Eddy-viscosity and the rate of turbulent kinetic energy dissipation (k-$\varepsilon$) where turbulence is fully parameterized and thus cannot provide information about turbulence structures and its consequent effects on the atmospheric chemistry (Meroney et al., 1995, 1996; Li et al., 2008). Some of these RANS models are able to resolve buildings and trees e.g. MITRAS
(the microscale obstacle-resolving transport and stream model), (Salim et al., 2018) and ASMUS (A numerical model for simulations of wind and pollutant dispersion around individual buildings), (Gross, 1997), however, due to their inherent weakness of parameterizing flow, RANS models are less accurate (Xie and Castro, 2006; Blocken, 2018; Maronga et al., 2019). Dispersion of gaseous species is essentially unsteady and cannot be predicted by steady-state approach, therefore, we

need turbulence resolving simulations to explicitly resolve unsteadiness and intermittency in the turbulent flow (Chang and Meroney, 2003).

In contrast to RANS, Large eddy simulation (LES) models are able to resolve turbulence and provide detailed information on the relevant flow variables (Baker et al., 2004; Li et al., 2008; Maronga et al., 2015, 2020). A large number of turbulence-resolving LES models are being used to investigate urban processes at scales from boundary layer to street canyons, e.g. Henn and Sykes (1992); Walton et al. (2002); Walton and Cheng (2002); Chang and Meroney (2003); Baker et al. (2004); Chung and Liu (2012); Nakayama et al. (2014). Many large eddy simulations studies that include transport of reactive scalars have been conducted e.g.,Baker et al. (2004), modelled the NO-NO$_2$-O$_3$ chemistry and dispersion in an idealised street canyon and Vila-Guerau de Arellano et al. (2006) investigated the influence of shallow cumulus clouds on the pollutant transport and transformation by means of LES. With the increasing computational power, more chemical reactants and mechanisms have been added into LES codes, for example the formation of ammonium nitrate (NH$_4$NO$_3$) aerosol including dry deposition (Barbaro et al., 2015) and photostationary equilibrium (Grylls et al., 2019). The NCAR LES model with coupled MOZART2.2 chemistry (Kim et al., 2012) includes a quite detailed description of isoprene oxidation and its products. This model was also applied by Li et al. (2016) in order to investigate turbulence-driven segregation of isoprene over a forest area. Furthermore, Vilà-Guerau De Arellano and Duynkerke (1997), Vilà-Guerau de Arellano et al. (2004a), Vilà-Guerau de Arellano et al. (2004b), Górska et al. (2006), Ouwersloot et al. (2011), Lenschow et al. (2016), and Lo and Ngan (2017) investigated the vertical turbulent transport of trace gases in the convective planetary boundary layer. Most of the LES-based pollutant dispersion studies investigated the flow and ventilation characteristics in street canyons (Liu et al., 2002; Walton et al., 2002; Walton and Cheng, 2002; Baker et al., 2004; Cui et al., 2004; Li et al., 2008; Moonen et al., 2013; Keck et al., 2014; Toja-Silva et al., 2017) or other idealised structures. These studies indicated that LES coupled air pollution models can help to explain microscale urban features and observed pollutant transport characteristics in cities (Han et al., 2019). However, most of these LES models either do not contain detailed atmospheric composition or full range of urban climate features such as human biometeorology, indoor climate, thermal stress, a detailed air chemistry or these are difficult to adapt to the state-of-the-art parallel computer systems due to lack of scalability on clustered computer systems which restrict their applicability on large domains (Maronga et al., 2015, 2019).

This paper describes the chemistry model that has been implemented in the PALM model system 6.0 as part of PALM-4U (PALM for urban applications) components. In the past PALM has been used to study urban turbulence structures (Letzel et al., 2008). Some studies also investigated dispersion of reactive pollutants (NO,NO$_2$ and O$_3$) using simple steady-state chemistry in PALM in the urban street canyons (Cheng and Liu, 2011; Park et al., 2012; Han et al., 2018, 2019). The PALM-4U components are essentially designed for urban applications and offers several features required to simulate urban environment such as an energy balance solver for urban and natural surfaces, radiative transfer in the urban canopy layer, biometeorological analysis products, self-nesting to allow very high resolution in regions of special interest, atmospheric aerosols and gas-phase chemistry (Raasch and Schröter, 2001; Maronga et al., 2015, 2020).

In order to offer the latter feature, an 'online' coupled chemistry model has been implemented in the PALM Model, which is presented in this paper. The chemistry model includes chemical transformations in the gas phase, a simple photolysis pa-

rameterization, dry deposition processes and an emission module to read anthropogenic pollutant emissions. The gas-phase chemistry has been implemented using the Kinetic PreProcessor (KPP) (Damian et al., 2002) allowing automatic generation of the corresponding model code in order to obtain the necessary flexibility in the choice of chemical mechanisms. Due to the very high computational demands of an LES-based urban climate model, this flexibility with respect to the degree of detail of the gas-phase chemistry mechanism is of critical importance. A number of ready-to-use chemical mechanisms with varying complexity and detail are supplied with PALM. Furthermore, the gas-phase chemistry is coupled to the aerosol module SALSA (A sectional aerosol module for large scale applications), (Kokkola et al., 2008) implemented in PALM (Kurppa et al., 2019) which includes a detailed description of the aerosol number size distribution, chemical composition and aerosol dynamic processes.

The analysis provided in this work is mostly qualitative and intended to show first applications of the chemistry model in a real urban environment, thereby demonstrating its capabilities and its flexibility. A detailed description of the chemistry model and its implementation to PALM is provided in section 2. The model application and details of the numerical setup for a case study representing a selected area in central Berlin, Germany, are described in section 3, whereas results of the application of the chemistry model and comparison of simulation results with observations are provided in section 4. In the end, concluding remarks are provided in section 5.

## 2 Model Description

### 2.1 PALM and PALM-4U

The PALM model system 6.0, consists of the PALM core and PALM-4U (PALM for urban applications) components (Maronga et al., 2020) that have been added to PALM under the MOSAIK (Model-based city planning and application in climate change) project (Maronga et al., 2019), one of which is the chemistry module described in this paper. PALM solves the non-hydrostatic, filtered, incompressible Navier-Stokes equations on a Cartesian grid in Boussinesq-approximated form for up to seven prognostic variables: the three velocity components ($u$, $v$, $w$) on staggered Arakawa C grid, and four scalar variables namely potential temperature ($\theta$), water vapour mixing ratio ($q_v$), a passive scalar $s$ and the subgrid-scale turbulent kinetic energy (SGS-TKE) $e$ (in LES mode) (Maronga et al., 2019, 2020).

By default, these SGS terms are parameterized using a 1.5-order closure after (Deardorff, 1980). The model uses a fifth-order advection scheme of Wicker and Skamarock (2002) and a third order Runge-Kutta scheme for time-stepping. Monin-Obukhov similarity (MOST) is assumed between every individual surface element and the first computational grid level. For details on meteorological and urban climate features and available parameter options, see Resler et al. (2017), and Maronga et al. (2020). Additionally, PALM includes options of fully interactive surface and radiation schemes, and a turbulence closure based on the Reynolds-averaged Navier-Stokes equations (RANS) mode. Details of the dynamic core of the model are described in Maronga et al. (2015) and Maronga et al. (2020).

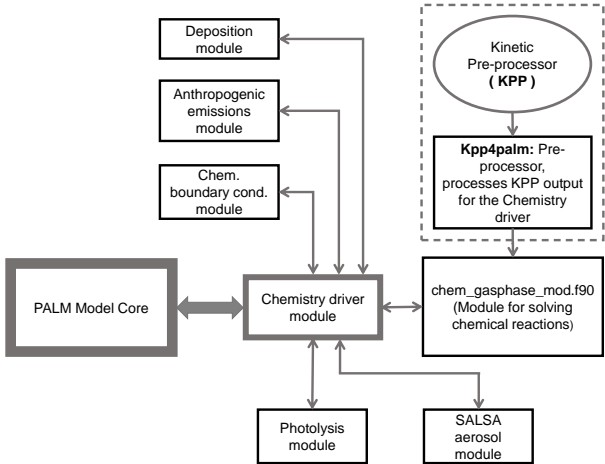

**Figure 1.** Schematic representation of the chemistry model (PALM-4U component) of the PALM model system. The arrows show interaction between the PALM model core, the chemistry driver module and sub-modules. The dashed box indicates the chemical preprocessor which generates subroutines to solve chemical reactions.

## 2.2 The chemistry model

Atmospheric chemistry is integrated into the PALM code as a separate module (Fig. 1) that utilizes the meteorological fields of PALM as input. Chemistry is coupled 'online' with the PALM model, i.e., the prognostic equations for the chemistry compounds are solved consistently with the equations for momentum, heat, and water constituents. This implementation of chemistry allows for a future consideration of the impact of trace gases and aerosol particles on meteorology by radiative effects and aerosol cloud interactions. As shown in Fig. 1 the main chemistry driver module calls and exchanges data with separate modules for the chemistry solver, photolysis, handling of lateral boundary conditions, concentration changes due to emissions, and deposition.

Depending on the need, a user can select a chemistry mechanism of different complexity. The Fortran code for the selected gas-phase chemistry mechanism is generated by a preprocessor based on KPP (Damian et al., 2002). The latter is described in more detail in Section 2.2.2. Besides chemical transformations in the gas phase and a simple photolysis parameterization (Section 2.2.3) the chemistry module includes dry deposition (Section 2.2.5), an interface to the aerosol module SALSA (Kurppa et al., 2019) (Section 2.2.4) and an option for anthropogenic emissions (Section 2.3).

### 2.2.1 Prognostic equations

When gas phase chemistry is invoked, $N$ additional prognostic equations are solved, with $N$ being the number of variable compounds of the chemical reaction scheme. Except for the SGS flux terms, the overbar indicating filtered quantities is omitted to improve readability. The three-dimensional prognostic equation for an atmospheric pollutant then read as:

$$\frac{\partial c_n}{\partial t} = -\frac{1}{\rho}\frac{\partial \rho\, u_j\, c_n}{\partial x_j} - \frac{1}{\rho}\frac{\partial \rho\, \overline{u_j''\, c_n''}}{\partial x_j} + \left(\frac{\partial c_n}{\partial t}\right)_{\text{chem}} + \Psi_n, \quad \text{with} \quad i,j \in (1,2,3), \tag{1}$$

where $c_n$ $(n = 1, N)$ is the concentration of the respective air constituent, which can be either a reactive or passive gas-phase species or an aerosol particulate matter compound. The term on the left-hand side is the total time derivative of the pollutant concentration. The first two terms on the right-hand side represent the explicitly resolved and the SGS transport of the scalar chemical quantity in $x$, $y$ and $z$ direction. A double prime indicates a SGS variable. The third term represents the change in
concentration ($c_n$) of the trace gas $n$ over time due to production and loss to chemical reactions, which can be described as follows:

$$\left(\frac{\partial c_n}{\partial t}\right)_{\text{chem}} = \phi_n(c_{m\neq n}) + \varphi_n(c_{m\neq n})\cdot c_n, \tag{2}$$

where $\phi_n$ and $\varphi_n$ indicate the production and loss, respectively, of species $n$. For most of these production and loss reactions the rates are dependent on temperature and pressure. The last term in Eq. 1 ($\Psi_n$) stands for sources (i.e., emissions) and
sinks (i.e., deposition and scavenging). The number of prognostic equations depends on the number of species included in the chemical mechanism and it is determined automatically during the KPP preprocessing step (Section 2.2.2).

### 2.2.2 Gas phase chemistry implementation

The Fortran subroutines for solving the chemical reactions of a given gas phase chemistry mechanism are generated automatically with the Kinetic PreProcessor (KPP), version 2.2.3 (Damian et al., 2002; Sandu et al., 2003; Sandu and Sander, 2006).
KPP creates the code from a list of chemical reactions that represent a certain chemical mechanism. Within the PALM environment, the subroutines with the integrator for the desired gas-phase chemistry mechanism are generated by a preprocessor named kpp4palm, which is based on the KP4 preprocessor (Jöckel et al., 2010). As a first step, kpp4palm starts the KPP preprocessor. As a second step, the code from KPP is transformed to a PALM subroutine. As described by Jöckel et al. (2010), the preprocessing includes also an optimisation of the LU (lower–upper) decomposition of the sparse Jacobian of the ordinary
differential equation system for the chemistry rate equations.

KPP offers a variety of numerical solvers for the system of coupled ordinary differential equations describing the chemical reactions. Tests comparing the performance of the Rosenbrock solvers implemented in KPP have shown that the use of the most simple Rosenbrock solver, Ros-2, did not lead to significantly different results than the use of the Rosenbrock solvers

with higher order (Sandu and Sander, 2006; Jöckel et al., 2010). Therefore, the Ros-2 solver was chosen as the default solver for the PALM-4U chemistry model.

The automatic code generation by kpp4palm and KPP allows for high flexibility in the choice of gas phase chemical mechanisms and numerical solvers. Since the number of chemical compounds of a mechanism from KPP is used to determine the number of prognostic equations (Eq. 1), it is also possible to add prognostic equations for an arbitrary number of passive tracers by simply including reactions of the form $A \rightarrow A$ in the list of chemical reactions which serves as input for KPP. For example, the passive tracer mechanism "passive" contains the following equations in the passive.eqn file:

{ 1.} PM10 = PM10 :                1.0 ;

{ 2.} PM25 = PM25 :                1.0 ;

(see Sect. S2..S9 of the Supplement for all .eqn files).

The chemistry model includes a number of ready-to-use chemical mechanisms summarised in Table 1. The first two mechanisms describe only one or two passive tracers which represent $PM_{10}$ and $PM_{2.5}$ (particulate matter with the aerodynamic diameter $\leq 10\,\mu m$ and $\leq 2.5\,\mu m$, respectively) without any chemical transformations. As a representative of a 'full' gas-phase mechanism, the well-known Carbon-Bond-IV (CBM4) (Gery et al., 1989) is included. Although CBM4 has been replaced by the more detailed CB5 and CB6 mechanisms in the meantime, it is still applied in some models. The CBM4 mechanism was implemented in the PALM model, since – with 32 compounds – it is the smallest of the full mechanisms. Nevertheless, the comparatively large number of species precludes the use of the CBM4 mechanism for practical applications over larger domains. Therefore, we also included computationally less demanding mechanisms, such as the SMOG mechanism and its simplified version, the 'SIMPLE' mechanism. By large, the photostationary equilibrium ( PHSTAT mechanism) represents the most simple mechanism consisting of only three species and two reactions. The latter two mechanisms are also supplied with an additional passive tracer which can be used to represent $PM_{10}$ (PHSTATP and SIMPLEP mechanisms). Three more mechanisms which can be used in combination with the sectional aerosol module SALSA (Kurppa et al., 2019) are described in Section 2.2.4.

Two of the currently available mechanisms, SMOG and SIMPLE, include only major pollutants such as ozone ($O_3$), nitric oxide (NO), nitrogen dioxide ($NO_2$), carbon monoxide (CO), a highly simplified chemistry of volatile organic compounds (VOCs) and a very small number of products. For the convenience of the users, it is not required to run kpp4palm for the ready-to-use mechanisms (Table 1) as their Fortran subroutines are already supplied with PALM. Currently, PHSTATP is the default mechanism which will automatically be compiled with the rest of the PALM source code when the chemistry option is switched on. However, users can also add modified versions of the existing chemical mechanisms or define completely new mechanisms according to their specific needs.

### 2.2.3 Photolysis frequencies

The parameterization of the photolysis frequencies is adopted from the Master Chemical Mechanism (MCM) v3 according to Saunders et al. (2003). Photolysis frequencies are described as a function of the solar zenith angle $\vartheta$ and three parameters,

**Table 1.** Description of built-in chemical mechanisms

| No | Mechanism | Var. Species | Fixed Species | KPP Reactions | Real reactions | Photolysis |
|----|-----------|--------------|---------------|---------------|----------------|------------|
| 1 | PASSIVE1 | 1 | – | 1* | 0 | 0 |
| 2 | PASSIVE | 2 | – | 2** | 0 | 0 |
| 3 | PHSTAT | 3 | – | 2 | 2 | 1 |
| 4 | PHSTATP | 4 | – | 3* | 2 | 1 |
| 5 | SIMPLE | 9 | water vapour | 7 | 7 | 2 |
| 6 | SIMPLEP | 10 | water vapour | 8* | 7 | 2 |
| 7 | SMOG | 13 | water vapour, $O_2$, $CO_2$ | 12 | 12 | 2 |
| 8 | CBM4 | 32 | water vapour | 81 | 81 | 11 |

\* Includes one passive compound which is realised in the KPP environment by the 'reaction' $PM_{10} \rightarrow PM_{10}$

\*\* Includes passive compound which are realised in the KPP environment by the 'reactions' $PM_{10} \rightarrow PM_{10}$ and $PM_{2.5} \rightarrow PM_{2.5}$

which are specific for each photolysis reaction:

$$J = l\left(\cos\vartheta\right)^m \exp\left(-n\sec\vartheta\right). \tag{3}$$

Values for $l$, $m$ and $n$ are given for the relevant photolysis reactions in Saunders et al. (2003) and on the MCM web page (http://mcm.leeds.ac.uk/MCM/parameters/photolysis_param.htt). Currently only a simple parameterized photolysis scheme is
available for photochemical reactions. More extensive photolysis schemes such as the Fast-J photolysis scheme (Wild et al., 2000) that are based on the radiative transfer modelling will be included in the future. These models will also make use of the shading due to buildings, which is already implemented for the shortwave radiation in the PALM-4U urban surface model but so far not for the simple photolysis scheme.

### 2.2.4   Coupling to SALSA aerosol module

The sectional aerosol module SALSA2.0 (Kokkola et al., 2008) has recently been implemented into PALM and a detailed description is given in Kurppa et al. (2019). SALSA describes the aerosol number size distribution, aerosol chemical composition and aerosol dynamic processes. Currently, the full SALSA implementation in PALM includes the following chemical compounds in the particulate phase: sulphate ($SO_4^{2+}$), organic carbon (OC), black carbon (BC), nitrate ($NO_3^-$), ammonium ($NH_4^+$), sea salt, dust and water ($H_2O$). Aerosol particles can grow by condensation and dissolution to liquid water of gaseous
sulphuric acid ($H_2SO_4$), nitric acid ($HNO_3$), ammonia ($NH_3$) and semi- and non-volatile organics (SVOC and NVOC), which establishes a link between SALSA and the chemistry module.

SALSA is coupled to the gas-phase chemistry when the gas-phase compounds listed above ($H_2SO_4$, $HNO_3$, $NH_3$, SVOC and NVOC) are either included in the gas phase chemistry scheme or are derived from prognostic variables of the gas phase chemistry. Currently, PALM-4U includes three different mechanisms in which SALSA is coupled with the chemistry model. In
the mechanisms 'SALSAGAS' and 'SALSA+PHSTAT', $H_2SO_4$, $HNO_3$, $NH_3$, SVOC and NVOC are treated as passive compounds and are only transported within the gas-phase chemistry model, whereas in 'SALSA+SIMPLE', $HNO_3$ is formed by

the reaction $NO_2 + OH \rightarrow HNO_3$. Additionally, any of the other mechanisms given in Table 1 or any user-supplied mechanism can also be coupled to SALSA.

### 2.2.5 Deposition

Deposition is a major sink of atmospheric pollutant concentrations. Currently, only dry deposition processes are included as precipitation (leading to wet deposition of pollutants) is not yet included in PALM. For dry deposition, a resistance approach is taken where the exchange flux is the result of a concentration difference between atmosphere and earth surface and the resistance between them. Several pathways exist for this flux, each with its own resistance and concentration. The aerodynamic resistance depends mainly on the atmospheric stability. In PALM, it is calculated via MOST, based on roughness lengths for heat and momentum and the assumption of a constant flux layer between the surface and the first grid level.

For gases, the quasi-laminar layer resistance depends on the atmospheric conditions and diffusivity of the deposited gas and it is calculated following Simpson et al. (2003). Finally, the surface (canopy) resistance for gases, which is the most challenging resistance to estimate due to the enormous diversity of surfaces, is calculated using the DEPAC module (Van Zanten et al., 2010). DEPAC is widely used in the flux modelling community (e.g. (Manders et al., 2017; Sauter et al., 2018). The surface resistance parameterizations are different for different land use types defined in the model. In DEPAC, three deposition pathways for the surface resistance are taken into account:

- through the stomata

- through the external leaf surface

- through the soil

DEPAC is extensively described in a technical report by Van Zanten et al. (2010). It also includes a compensation point for ammonia which is currently set to zero in PALM.

For the passive particulate matter in the chemistry model, the land-use dependent deposition scheme of Zhang et al. (2001) has been implemented into PALM. The formulations have been chosen as they include an explicit dependence on aerosol size. For particulate matter, the deposition velocity is calculated by the gravitational settling or sedimentation velocity (mainly relevant for the larger particles), the aerodynamic resistance (see above) and the surface resistance. The sedimentation velocity mainly depends on particle properties, the gravitational acceleration and the viscosity coefficient of air. The formulation for the surface resistance is empirical with parameters that are based on a few field studies including the collection efficiencies for Brownian diffusion, impaction and interception, respectively, and a correction factor representing the fraction of particles that stick to the surface depending on the surface wetness. Further details can be found in Zhang et al. (2001).

### 2.3 Traffic emissions

The chemistry model of PALM-4U includes a module for reading gaseous and passive anthropogenic emission input from traffic sources and converting it to the appropriate format. These emission data can be provided in three possible levels of detail

(LODs), depending on the amount of information available at the user's disposal. With LOD 0, ("PARAMETERIZED", mode), traffic emissions are parameterized based on emission factors specific to particular street types. All street segments contained in the domain will be classified into "main" and "side" street segments. A mean surface emission flux tendency for each chemical species contained the active chemical mechanism, in kilograms per square meter per day, will be provided together with a weighting factor for main and side street emissions in the PALM parameter file. Street type classification based on Open-StreetMap definitions (OpenStreetMap contributors, 2017) is to be included in the PALM static driver (Maronga et al., 2020). A diurnal profile derived from traffic counts is implemented to disaggregate total emissions into hourly intervals. Currently a default profile is applied to all species for main and side street segments, and details can be found in the online documentation for the PALM-4U chemistry model (https://palm.muk.uni-hannover.de/trac/wiki/doc). Future plans include expansion of the LOD 0 emission model to accommodate further modes of anthropogenic emissions such as domestic heating. More detailed traffic emission data can be provided in gridded form in PALM-specific NetCDF files (Maronga et al., 2020). LOD 1 emissions are gridded annual emission data for each sector (e.g. industry, domestic heating, traffic), which will be temporally disaggregated using sector-specific standard time factors. With LOD 2, the user can introduce preprocessed gridded emission data that are already temporally disaggregated, e.g. in hourly intervals.

## 2.4 Initial and boundary conditions

Lateral boundary conditions for chemical compounds can be chosen in the same way as the lateral boundary condition for other scalars, e.g. potential temperature, being either cyclic conditions or non-cyclic (Maronga et al., 2020). In most urban applications, chemistry requires non-cyclic boundary conditions, because cyclic conditions lead to accumulation of pollutants to the modelling domain if pollutant emissions exist. As part of the PALM-4U components, nesting has been implemented to the chemistry module. In offline nesting, PALM can be coupled to a larger (meso, regional, or global) scale model to provide dynamic boundary conditions for the meteorological variables as well as air pollutants. As larger scale models do not fully resolve turbulence, a synthetic turbulence inflow generator has been introduced (Gronemeier et al., 2015).

Initial concentrations of primary compounds to control the chemistry model are controlled by the chemistry namelist "&chemistry_parameters" in PALM parameter file. Options for prescribing initial conditions are available for both surface initial conditions and initial vertical profiles for the area or region of interest. There are no default initial concentrations and the user is responsible for providing these values based on, e.g., measured background concentration of primary chemical compounds for whom initial concentration are defined. These primary compounds must be part of the applied chemical mechanism.

## 3 Chemistry model application

In order to demonstrate the ability of the chemistry implementation, we performed simulations for an entire daily cycle in a realistic urban environment and compared simulation results against observational data. To analyse the effect of different chemical mechanisms with different complexity on the resulting concentrations, we performed simulations with three chemical mechanisms and one passive case where only transport and dry deposition was considered.

## 3.1 Modelled episode and modelling domain

The model was run for 24 hours from 00:00 UTC to 24:00 UTC for July 17, 2017 (0200 Central European Summer Time (CEST) 17 July 2017 to 02:00 CEST 18 July 2017) for a city quarter located around the Ernst-Reuter-Platz in Berlin, Germany. This particular day was chosen as it represents an 'ideal' Berlin summer day with mostly clear sky, some scattered clouds in the morning after a partly cloudy night, and only a few passing clouds in the afternoon. The temperature ranged between 289 K to 298 K with moderate winds predominantly from westerly direction. July 17, 2017 was a Monday, therefore, the diurnal cycle of the traffic emissions can be described by a typical weekday with relative maxima during the morning and evening rush hours.

Figure 2 shows the computational domain that covers an area of 6.71 km x 6.71 km (671 x 671 grid points) with a model top at 3.6 km above the surface. The horizontal grid spacing in $x$ and $y$ direction is 10 m. In the vertical, with 312 layers, the grid spacing is 10 m up to 2700 m, above which it increases by an expansion factor of 1.033 until the grid spacing in the $z$-direction reaches 40.0 m.

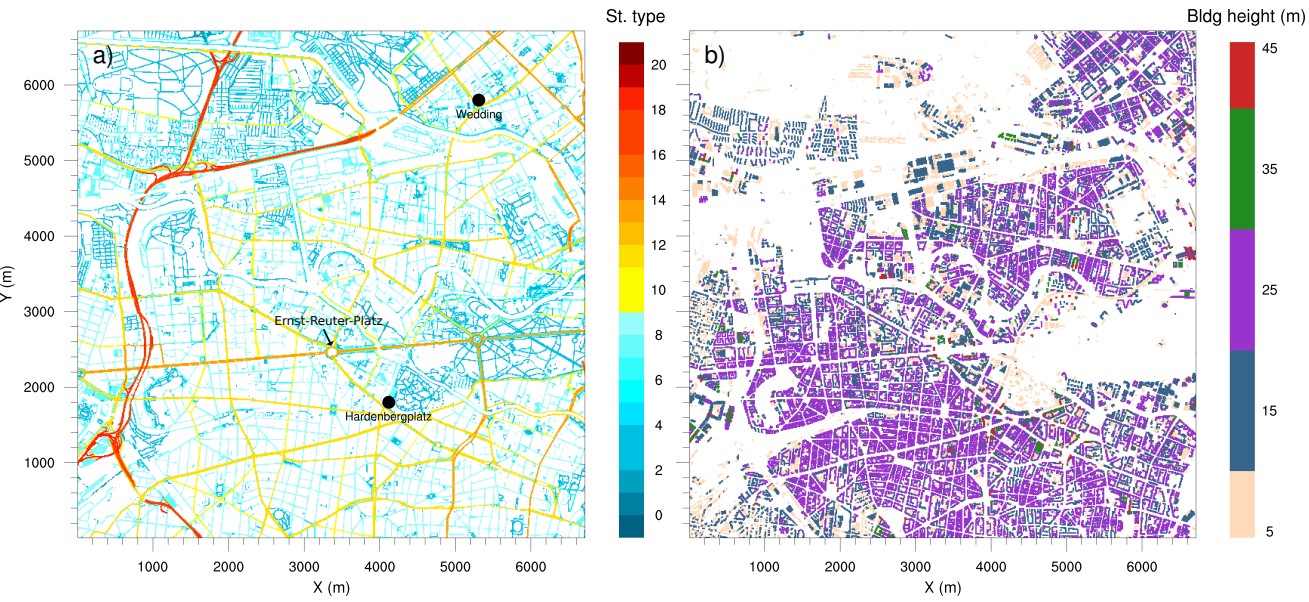

**Figure 2.** Simulation domain; left: street type, right: building height. The black dots show the location of the observational monitoring stations Wedding and Hardenbergplatz.

The topographic data with streets, buildings, water bodies, vegetation and other urban land surface features at a 10 m resolution has been processed for Berlin by German Space Agency (DLR) (Heldens et al., 2020). The street types shown in Fig. 2a are based on OpenStreetMap (OpenStreetMap contributors, 2017). The model domain includes 13 types of one-way and two-way roads. The building height data is based on CityGML data from FIS Broker Berlin (Senatsverwaltung für Stadtentwicklung und Wohnen, 2020, last accessed on 27.11.2020), Fig. 2b. With the exception of 10 buildings with a height

between 45 m and 65 m and two buildings, which are higher than 70 m, the building heights in the simulation domain are between 5 and 45 m.

The simulation domain contains five types of vegetation categories that include grass and shrubs of different height. Trees resolved by the canopy model are characterised by the three-dimensional leaf area density per unit volume (LAD). For the model configuration used here, LAD is considered for, i.e. up to a maximum height of 40 m above the ground and assumes values up to 3.1 m$^2$ m$^{-3}$ with an average value of 0.44 m$^2$ m$^{-3}$. Lake, river, pond and fountain categories are included in the domain for water type. Considering the urban surface, soil type of the entire domain is classified as "coarse" whereas pavement types in the domain are defined with six different categories of asphalt, concrete and stones.

## 3.2 Observational data

In addition to routine observations of near surface temperature, cloud cover, and wind speed and direction at the Berlin Tegel airport from the open access Climate Data Center (CDC) of the German Weather Service (Deutscher Wetterdienst, 2020, last accessed on 30.07.2020), we also analysed radiosonde data from Lindenberg (Oolman, 2017, last accessed on 30.07.2020), and aerosol backscatter observations from a ceilometer for 16 and 17 July 2017 to understand the vertical structure of the atmosphere in the area of interest. Ceilometer observations of aerosol back-scatter profiles were performed at the roof of the Charlottenburg building of the Technical University of Berlin (52.5123°N, 13.3279°E) as part of the Urban Climate Observatory (UCO) operated by the Chair of Climatology at Technische Universität Berlin for long-term observations of atmospheric processes in cities (Scherer et al., 2019a; Wiegner et al., 2002) and contribute to the research program Urban Climate Under Change [UC]$^2$ (Scherer et al., 2019b). Mixing layer heights were derived from the aerosol backscatter signal according to Geiß et al. (2017).

Observations from radiosonde and ceilometer indicate strong showers during the previous days that resulted in a very well mixed (almost moist adiabatic) layer in the lower troposphere that lead to an almost constant potential temperature gradient above the inversion, therefore, there was no residual layer at mid-night on 17 July, when the model was initialised for a 24 hour run. The ceilometer observations for 17 July 2017, also did not show disturbances by clouds, the mixed layer top, however, remained below 2000 m throughout the diurnal cycle.

The air quality measurements that are compared with the simulations are conducted at automated stations of the so-called BLUME network of the Berlin Department for the Environment, Transport and Climate Protection (Senate-Berlin, 2017) (last access: 14 August 2017). Two stations, Wedding and Hardenbergplatz, (Fig. 2) are located within the model domain. The average height of the air quality sensors at both stations is 4 m above ground.

The roadside air quality station at Hardenbergplatz is located at a busy junction with high traffic flow and in close proximity to the train and bus station. The station records only NO, and NO$_2$. The height of the buildings in the vicinity of the station ranges from 10 to 30 m. The dense road network of small and big streets to the north, south and west of the station warrants transport of traffic-related pollutants towards Hardenbergplatz when the station is located downwind of these directions. To the north-east of the station is the Berlin Zoo and Tiergarten park that spread over 5.2 km$^2$. Thus, with a NE flow, the Hardenbergplatz

air quality station would be downwind of the large vegetated area and less anthropogenic pollutants would be advected to the station, although the concentration of BVOC is expected to be larger.

The background air quality station of Wedding is located to the north in the outskirts of the city away from heavy traffic flow. Wedding air quality station records $PM_{10}$, NO, $NO_2$ and $O_3$. The city centre is located to the south of the station, so the station would likely record higher levels of $NO_x$ and $PM_{10}$ during southerly flow. Both stations collect data every 5 minutes which are then averaged to hourly data and made available on the Senate department web pages (Senate-Berlin, 2017).

### 3.3 Model configuration and initialisation

The PALM model system, version 6.0, revision 4450 and 4601 (only for flux profiles of chemical compounds), have been used in this study. A multigrid scheme has been used to calculate pressure perturbations in the prognostic equations for momentum (Maronga et al., 2015), and a third-order Runge-Kutta scheme (Williamson, 1980) has been used for time integration. The advection of momentum and scalars was discretized by a fifth-order advection scheme by Wicker and Skamarock (Wicker and Skamarock, 2002). Following Skamarock (2006); Skamarock and Klemp (2008), we employed a monotonic limiter for the advection of chemical species along the vertical direction in order to avoid unrealistically high concentrations within the poorly resolved cavities (e.g. courtyards represented by only a few grid points) which can occasionally occur due to stationary numerical oscillations near buildings. Rayleigh damping has been used above 2500 m in order to weaken the effect of gravity waves above the boundary-layer top.

The Rapid Radiative Transfer Model (RRTMG) (Clough et al., 2005), which is included in PALM, has been used to calculate radiation fluxes and radiation heating rates. Natural-type surfaces are treated by the land-surface model of PALM, while building surfaces are treated by the urban-surface model (Resler et al., 2017; Maronga et al., 2020). The surface roughness length is set according to the given building, vegetation, pavement and water types and based on the information from the static driver. The MOST then provides surface fluxes of momentum (shear stress), and scalar quantities (heat, moisture) at the lower boundary condition. The application of MOST assumption on urban surfaces has not been thoroughly evaluated. However, similar studies, for example Letzel et al. (2008) and Gronemeier et al. (2020), show that LES results were in good agreement with wind-tunnel data representing an urban setting. Based on these findings, it is assumed that MOST is applicable in the simulated urban surface.

Three chemical mechanisms namely PHSTAT, SMOG and CBM4 along with one no-reaction case have been applied (Table 1). The CBM4 (Gery et al., 1989) mechanism is the most complex mechanism currently included in PALM-4U. It includes VOC and $HO_x$ (hydrogen oxide radicals), chemistry and formation of ozone and further photochemical products. Assuming that the CBM4 is more accurate than the more simple mechanisms due to its more complete representation of atmospheric chemistry, the baseline simulation of this study was performed with CBM4. The SMOG photochemical mechanism was included for comparison as it contains a strongly simplified $NO_x$-$HO_x$-VOC chemistry with VOC just described by one single representative compound (Table 1). Due to a smaller number of species and reactions, the SMOG mechanism is much faster compared to the CBM4 mechanism. The objective to include the SMOG mechanism was to assess computational efficiency at the cost of accuracy of the description of the VOC chemistry. The most simple mechanism, PHSTAT, describes only the pho-

tostationary equilibrium between NO, $NO_2$ and $O_3$ and does not include any VOC chemistry or formation of any secondary compounds besides ozone. In a fourth experiment, the photostationary mechanism (PHSTAT) was applied with the gas-phase chemistry turned off, i.e. the chemical compounds were treated as passive tracers and only transport and dry deposition were allowed.

Only traffic emissions, which are parameterized depending on the street type ('PARAMETERIZED' option, LOD=0) were considered. Since the area of interest is in the inner part of the city with many major roads and domestic heating emissions are neglectable in July, this restriction seems justified. The traffic emissions of the 'PARAMETERIZED' option are based on emission factors derived from HBEFA (HandBook Emission FActors road transport; Hausberger and Matzer, 2017) and traffic counts provided by the Senate Department for the Environment, Transport and Climate Protection of Berlin. A mean surface

emission of 4745 $\mu$mol m$^{-2}$ day$^{-1}$ and 1326 $\mu$mol m$^{-2}$ day$^{-1}$ are applied for NO and $NO_2$, respectively, and weights of 1.667 for main streets and 0.334 for side streets are applied for the current study. Considering the diurnal cycle of emissions a typical temporal profile of traffic emissions (see section S1 of the supplement), with maxima at 8:00 CEST and 18:00–19:00 CEST based on traffic counts at Ernst-Reuter-Platz was applied.

As initial conditions profiles of $\theta$, $q$, $u$, $v$ and $w$, soil moisture and soil temperature were obtained from the output of the

operational mesoscale weather prediction model COSMO-DE/D2 (COnsortium for Small scale MOdelling, (Baldauf et al., 2011) for 17 July 2017. The INIFOR tool was used to prepare these initial conditions in a format that can be read by PALM (Kadasch et al., 2020). The initial profiles of pollutant concentration are based on the mean observed near surface concentrations of NO, $NO_2$ and $O_3$ from the stations of the BLUME network (section 3.2). Initial concentrations of NO, $NO_2$ and $O_3$ above 495 m were set to 0.0, 2.0 and 40.0 ppb respectively. Considering the strong impact of traffic emissions on local pollutant

concentrations, all grid points of the model domain were initialised with identical pollutant profiles.

At the lateral boundaries, cyclic boundary conditions are applied for the velocity components, $\theta$, $q$ and the chemical compounds. The application of cyclic boundary conditions may be justified by low variability in wind direction with prevailing westerly winds throughout the major part of 17 July 2017 and the large extent of the urban area upwind of the model domain. One main reason for not applying boundary conditions from COSMO-DE/D2 was that it requires a very large fetch to develop

turbulence, as turbulence quantities are not supplied by the boundary conditions from the mesoscale COSMO-DE/D2 simulation. Furthermore, chemistry fields for the lateral boundaries were not available from COSMO-DE/D2. Thus, cyclic boundary conditions were applied that ensure cyclic inflow of meteorological and chemistry variables that left the domain from the opposite lateral boundary. Therefore, a continuous inflow and outflow of pollutants in and out of the simulation domain is assumed. This is reasonable, as the simulation domain is located in the middle of a large highly urbanised area.

At the bottom boundary, a Dirichlet condition is applied to flow, $\theta$, and $q$ whereas a Neumann condition is applied to $e$, $p$ and chemical compounds. Moreover, a canopy drag coefficient $C_d = 0.3$ has been applied while the roughness is specified internally depending on vegetation type. At the top boundary, Dirichlet boundary conditions are applied to flow and $p$ only, initial gradient is applied to $\theta$ while Neumann boundary conditions are applied to $q$ and chemical compounds.

## 4 Results and discussion

The results of the chemistry model simulations are presented for 20 hours from 0300 UTC (0500 CEST ) to 2300 UTC (0100 CEST 18 July 2017). All plots and data in this case study are presented in CEST. The simulation output was exported to file every 10 minutes as instantaneous values and every 30 minutes as temporal averages. Since observational data are available in hourly averaged data, we used the 30 minutes averaged model data for comparison with observations. For all other plots we used instantaneous data.

### 4.1 Meteorology

Figure 3 shows vertical profiles of potential temperature, mixing ratio, wind speed and wind direction over the diurnal cycle. The profiles of potential temperature indicate a vertically well-mixed boundary layer during daytime, evolving from approximately 500 meters at 7:00 CEST to more than 2300 m at 21:00 CEST, while in the evening hours the near-surface layer stabilises. The mixed-layer depth agrees fairly well with the observed values from ceilometer measurements (horizontal bars in Fig. 3a), except for the late afternoon and evening hours, where the modelled boundary-layer depth is over-predicted by up to 15 %. This can have various reasons, e.g. in our simulations we neglected larger-scale processes such as subsidence or mesoscale advection. The wind comes from westerly directions at a mean wind speeds of about 6–9 m s$^{-1}$ within the mixed layer.

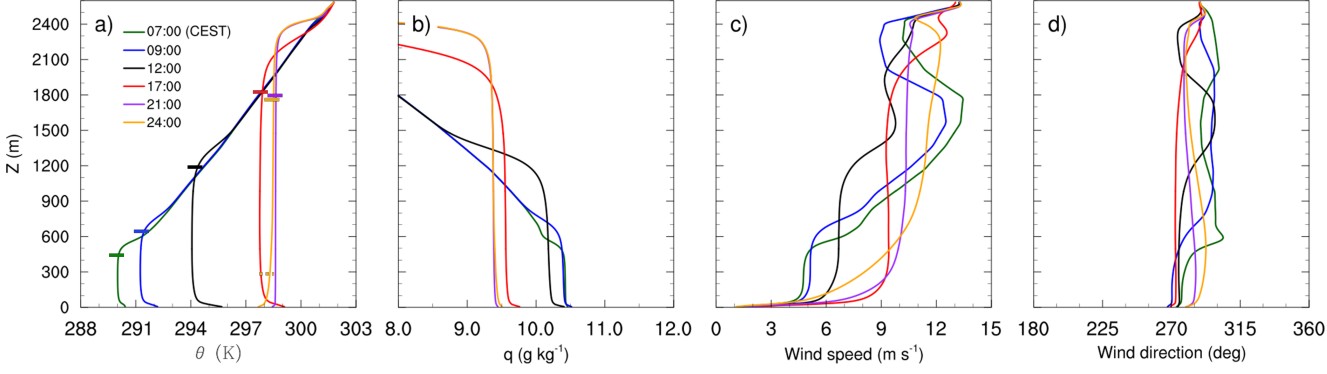

**Figure 3.** Vertical profiles of a) potential temperature, b) mixing ratio, c) wind speed and d) wind direction, at different times from morning to midnight on 17 July 2017. The horizontal bars in a) indicate the boundary-layer height derived from ceilometer observations.

### 4.2 Vertical mixing of NO$_2$ and O$_3$

Figure 4 shows mean profiles of concentrations and vertical fluxes of NO, NO$_2$, O$_3$ and CO for the selected times of the diurnal cycle on 17 July 2017, simulated with the CBM4 mechanism. Profiles and fluxes of CO are added to represent transport characteristics of passive species. Positive fluxes with a negative vertical gradient can be observed for NO and NO$_2$, indicating

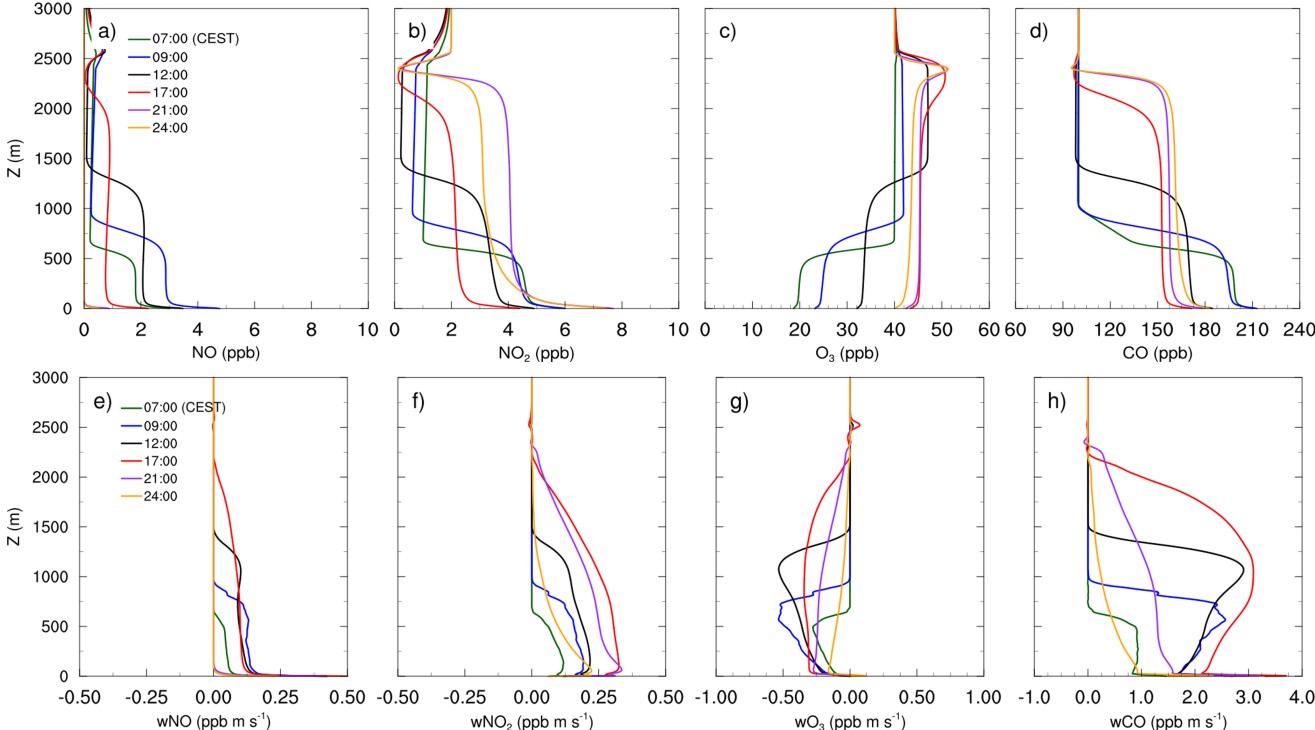

**Figure 4.** Mean vertical profiles of species concentration for a) NO, b) $NO_2$, c) $O_3$ and d) CO, as well as their total vertical fluxes (e-h) over the diurnal cycle on 17 July 2017, simulated with the CBM4 mechanism.

net upward transport of the respective compounds from the surface towards higher levels during the entire diurnal cycle. The emitted NO from traffic sources oxidised to $NO_2$ by reaction with the available $O_3$:

$$NO + O_3 \rightarrow NO_2 + O_2 \qquad (R1).$$

This leads to an increase of $NO_2$ and a decrease of $O_3$ within the boundary layer. As indicated by Fig. 4g) the growth of the
5  boundary layer after sunrise leads to downward transport of $O_3$ since ozone concentrations in the residual layer are higher than within the mixed layer. At the same time, photolysis of $NO_2$ (R2) and photochemical formation of $O_3$ (R3) by VOC chemistry retained and slightly increased the $O_3$ concentration within the boundary layer as well as in the residual layer.

$$NO_2 + h\nu \rightarrow NO + O(^3P) \qquad (R2),$$
$$O_2 + O(^3P) + M \rightarrow O_3 + M \qquad (R3).$$

10  Near surface concentrations of NO are 1 to 2 ppb higher than the remaining mixed layer throughout the diurnal course. At 9:00 CEST, owing to emissions from traffic sources, the NO concentration increased from 3 ppb to 5 ppb in the first few meters above the surface, then evenly dilutes in the shallow mixed layer and stabilises with a mixing ratio of 3 ppb. Besides the high $NO_x$ emissions in the morning hours, this increase can also be attributed to the onset of $NO_2$ photolysis (R2). During the rest of the day, NO concentrations gradually reduces, mostly due to dilution by vertical mixing during further growth of the

mixed layer as well as the fast reaction with $O_3$ (R1). The NO concentrations above the inversion are much lower and mostly influenced by the background levels thus making NO vertical flux positive Fig. 4e showing upward transport of NO.

Unlike NO, the $NO_2$ concentration profiles in Fig. 4b show only small differences in the morning hours (7:00 and 9:00 CEST). In the afternoon (12:00 and 17:00 CEST) when the convection is stronger, the $NO_2$ concentration is the lowest due to
the combined effect of upward vertical mixing indicated by Fig. 4f and photolysis of $NO_2$ (R2).

Reactions R1 – R3 between NO, $NO_2$ and $O_3$ do not result in a net gain of $O_3$ unless additional $NO_2$ is supplied (e.g. primary $NO_2$ from traffic emissions) and $O_3$ is formed by reaction R2 and R3. However, OH radical chain reactions of VOCs result in the formation of excess $NO_2$ (R4 and R5) and thus $O(^3P)$ (R2), which results in to a net $O_3$ gain (Cao et al., 2019). In a schematic form, the formation of $NO_2$ due to VOC oxidation can be summarised by,

$\qquad OH + RH + O_2 \rightarrow RO_2 \qquad$ (R4),
$\qquad RO_2 + NO \rightarrow RO + NO_2 \qquad$ (R5),

where RH stands for any explicitly described or lumped non-methane hydrocarbon and $RO_2$ represents any organic peroxy radical.

In addition to reaction R3, the $O_3$ levels within the mixed layer are also replenished through down-welling from above the
inversion during the day which is evident from the negative flux profiles of $O_3$ Fig. 4g. As a result $O_3$ concentration gradually increased from 18 ppb at 07:00 CEST to 43 ppb at 17:00.

In the evening hours (21:00 hour onward), the NO concentration is reduced to 1 ppb near the surface while it is completely removed from the residual layer. In the reduced(no) solar radiation, $NO_2$ photolysis (R2) slows down (stops) whereas emission and limited R1 reaction (because of very low available NO) favours an increase in the $NO_2$ mixing ratio. This resulted in
the highest near surface $NO_2$ concentrations at 24:00 CEST in the nocturnal stable layer. In the well mixed residual layer above, the $NO_2$ concentration at 24:00 CEST is up to 1 ppb lower than $NO_2$ concentrations at 21:00 CEST. Ozone is also reduced up to 3 ppb at 24:00 CEST but remains well mixed in the residual layer above the shallow nocturnal stable boundary layer. Carbon monoxide has been added to the analysis as a proxy to passive species Fig. 4d . In the morning hours, the concentration is highest, then it gradually decreases due to turbulent mixing in the rapidly growing mixed layer and partly due
to dry deposition.

Figure 5 shows YZ vertical cross sections at Hardenbergplatz for $NO_2$ and $O_3$. Turbulence and the growth of the mixed layer height caused downward vertical mixing and $O_3$ entrainment near boundary layer top that resulted in increase of $O_3$ concentration as well as in decrease of $NO_2$ concentrations to 2.0 ppb. Near the ground, $NO_2$ concentration is still of the order of 10 ppb. At the top of the boundary layer, the concentration of OH and $HO_2$ were higher than in the surrounding areas,
which indicates active $NO_x$-VOC-$O_3$ chemistry in the turbulent entrainment zone.

## 4.3 Spatial distribution of pollutants

The spatial distribution of $NO_2$ and $O_3$ concentrations 5 m above surface (Fig. 6) are discussed for 9:00 and 21:00 CEST as simulated with the CBM4 mechanism. The concentrations of $NO_x$, aldehyde and hydrocarbon species (not shown), which are emitted by road traffic, were high in the morning and evening hours due to stable conditions and high emissions.

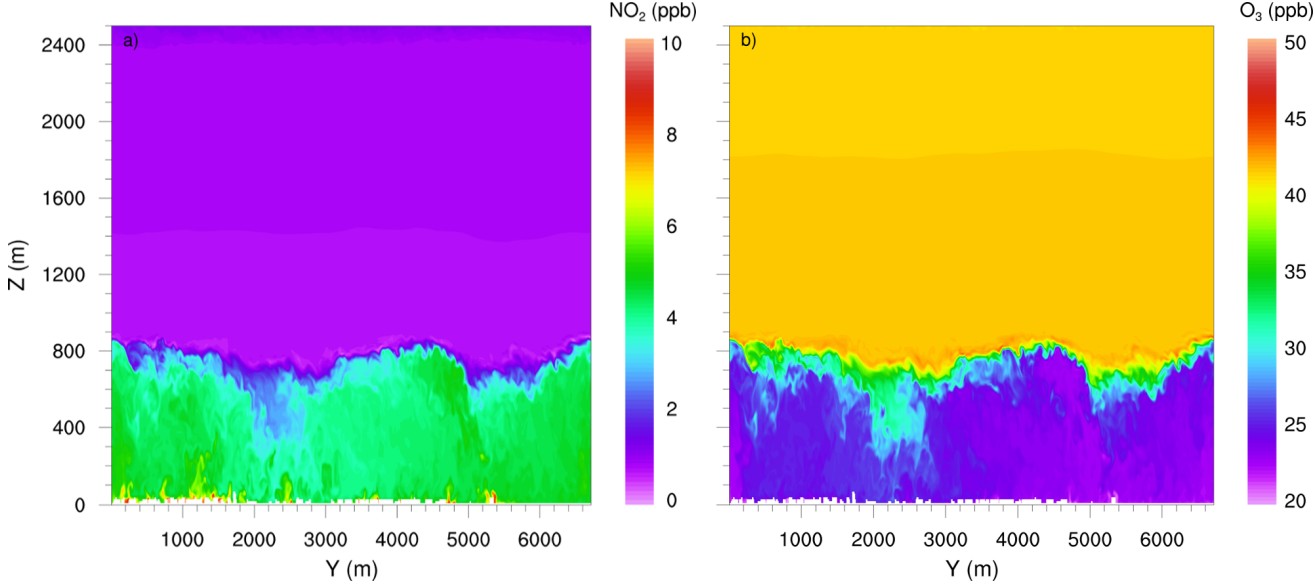

**Figure 5.** YZ vertical cross-sections of $NO_2$ and $O_3$ (drawn through Hardenbergplatz (Fig. 2a)), simulated with the CBM4 mechanism at 09:00 CEST on 17 July 2017.

Most of the high $NO_2$ concentrations in the morning hours (Fig. 6a) are predicted in the street canyons in the southern part of the simulation domain, where numerous buildings with a height of approximately 20 m are present (see Fig. 2a). In the street canyons of this area, wind speeds of less than 2 m s$^{-1}$ were simulated below 20 m, while wind speeds were almost twice as high over the open areas in the northern part of the model domain (not shown). Therefore, emitted $NO_x$ is more diluted by
5   advection over the open areas. Furthermore, in the morning hours compared to the open spaces, vertical mixing and transport in the street canyons is inhibited mainly due to delayed heating of the ground which is attributed to the shading of the surrounding buildings. Therefore, $O_3$ is predominantly titrated by reaction R1 (Fig. 6b) over the road network reducing its concentration to the order of 20 ppb, while over open spaces and vegetation (specially southern part of Tiergarten) a slightly higher $O_3$ mixing ratio (27 ppb) is found. The initial $O_3$ values near the surface were set to 10 ppb increasing to 40 ppb around 500 m above the
10  surface. Higher concentration of $O_3$ (30 ppb) near the surface in the open spaces indicates strong vertical mixing and downward transport of $O_3$ in the morning hours of the day. In the evening, however, the $NO_2$ distribution is somewhat more uniform over the entire road network (Fig. 6c). This is attributed to emissions from traffic with reduced or no photolysis of $NO_2$ (R2) and titration of $O_3$. Under the low-wind conditions below 50 m, ventilation is reduced and leads to an increase in $NO_2$ over street crossings, main and side roads. Consequently, the $NO_2$ concentration increased by 30 % in the evening hours over roads and
15  adjacent paved areas, whereas over the vegetated areas (grass, crops, shrubs and trees), $NO_2$ concentrations is around 5 ppb in the morning hours and reached up to 10 ppb in the evening hours. The daytime $NO_x$-$O_3$-VOC chemistry resulted in elevated $O_3$ levels which increased by more than 100 % as compared to the morning concentrations, mostly due to photochemical

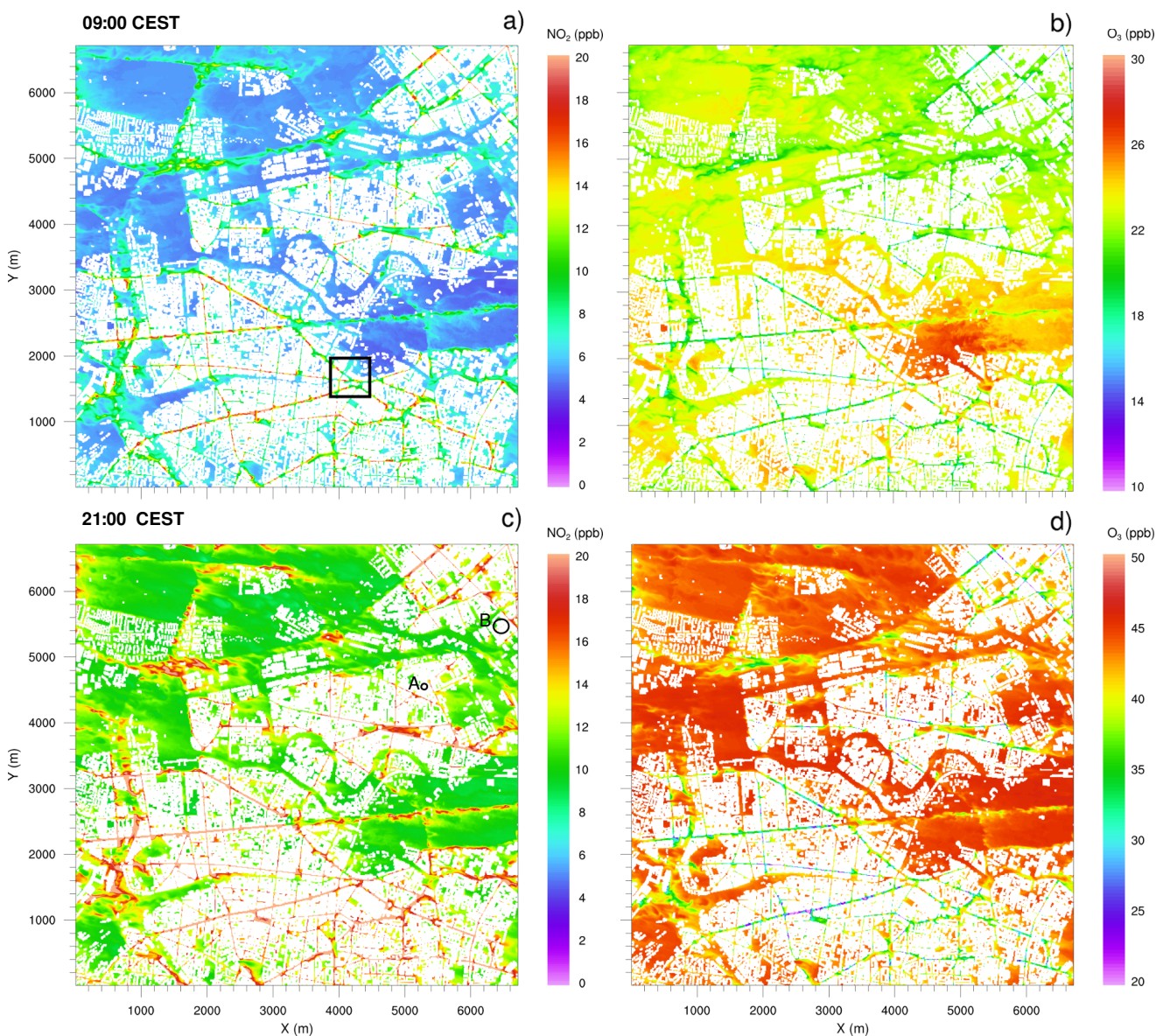

**Figure 6.** Horizontal cross-sections of near surface NO$_2$ and O$_3$ concentrations at 09:00 and 21:00 hours on 17 July 2017, simulated with the CBM4 mechanism. The black square in "a)" indicates the location of the small domain presented in Fig. 7. Locations A and B in plot "c" indicates isolated street canyons plotted in Fig. 8.

production of O$_3$ (Reaction R2 and R3) during daytime. In the evening hours (Fig. 6d), O$_3$ is largely titrated by reaction R1 over the road network. In certain sections of the street canyons where micro-meteorological conditions are more favourable,

O$_3$ levels decreased to 20 ppb. However, in the open spaces and over vegetation with very low NO, the O$_3$ concentration is of the order of 50 ppb.

To provide an overview of the pollutant dispersion at the street level, a small section of the model domain (500 x 500 m) in the Hardenbergplatz area has been analysed (Fig. 7). This small urban section is characterised by a typical urban environment with streets, paved areas, buildings with varying heights, and some vegetated areas of the Tiergarten located to the north-east. The dispersion of air pollutants in a street canyon generally depends on the aspect ratio (building height to street width ratio) and rate at which the street exchanges air vertically with the above roof-level atmosphere and laterally with connecting streets (N'Riain et al., 1998). Figure 7 shows the dispersion and chemical transformation of NO$_2$ and O$_3$ in the street canyons and surrounding areas. The vegetated area of Tiergarten north-east of the domain has relatively high O$_3$ levels and small NO$_2$ concentrations which is attributed to the low NO concentrations over the vegetation that lead to reduced O$_3$ titration (R1). The model has emissions only from traffic sources and therefore availability of the primary species other than roads depends upon the dispersion of the primary chemical compounds such as NO. In the street canyons, O$_3$ is titrated by NO by reaction R1 resulting in production of NO$_2$.

In the morning hours, NO$_2$ concentration increases in some sections of the street canyons (location A, B, C Fig. 7a) which are located to the west/southwest of the tall buildings and thus lie under the shade of the building structures. Due to the particular street geometry, and flow dynamics above the urban canopy, these sections also experience weak wind conditions. The elevated NO$_2$ concentrations are due to later onset of the turbulence and vertical exchange caused by reduced solar radiation in the shade of the buildings. This results in increased residence time of NO and O$_3$ due to very localised calm/near calm conditions in the street canyons. The daytime NO-O$_3$-VOC chemistry significantly increases O$_3$ concentration. In the evening hours when photolysis of NO$_2$ is again decreasing, the NO$_x$ and O$_3$ chemistry mostly controlled by reaction R1 and wind speed in the street canyons. The street sections (locations D,E,F in Fig. 7c) with low wind/calm conditions experience relatively elevated NO$_2$ and low O$_3$ levels as under calm conditions NO has enough residence time to titrate O$_3$ and thus contribute towards increased NO$_2$ concentrations. Streets with stronger winds over the open spaces (train tracks, Tiergarten area), experience lower than average NO$_2$ levels.

At some locations, PALM simulated unrealistically high NO$_2$ concentrations. Figure 8 shows time series plots along with $xy$ cross-section of the street canyons (in the inset) where the model simulated high NO$_2$ concentrations at location A and B as indicated in Fig. 6c. We selected four grid points at location A and randomly five grid points from the discontinued street canyon at location B. During the day, NO$_2$ concentrations remain up to 50 ppb, however, after 18:00 CEST, NO$_2$ concentrations increase rapidly at both locations. By 21:00 CEST, NO$_2$ concentration at location A is on the order of 75 ppb Fig. 8a, whereas, at location B, NO$_2$ levels ranges between 65 and 90 ppb (Fig. 8b). At 01:00 CEST, the NO$_2$ concentration at both locations A and B reaches 150 ppb and 275 ppb respectively.

Due to relatively coarse grid resolution of $10\,\mathrm{m}$, as applied in this study, street canyons are partly only resolved by one or two grid points and sometimes enclosed cavities occur. Within such geometries the flow is only poorly resolved by the numerical grid, which means that the bulk of the transport is parameterized by the subgrid model. However, as the subgrid TKE is also very low in such narrow canyons or cavities (not shown), the total vertical transport is rather low, especially in

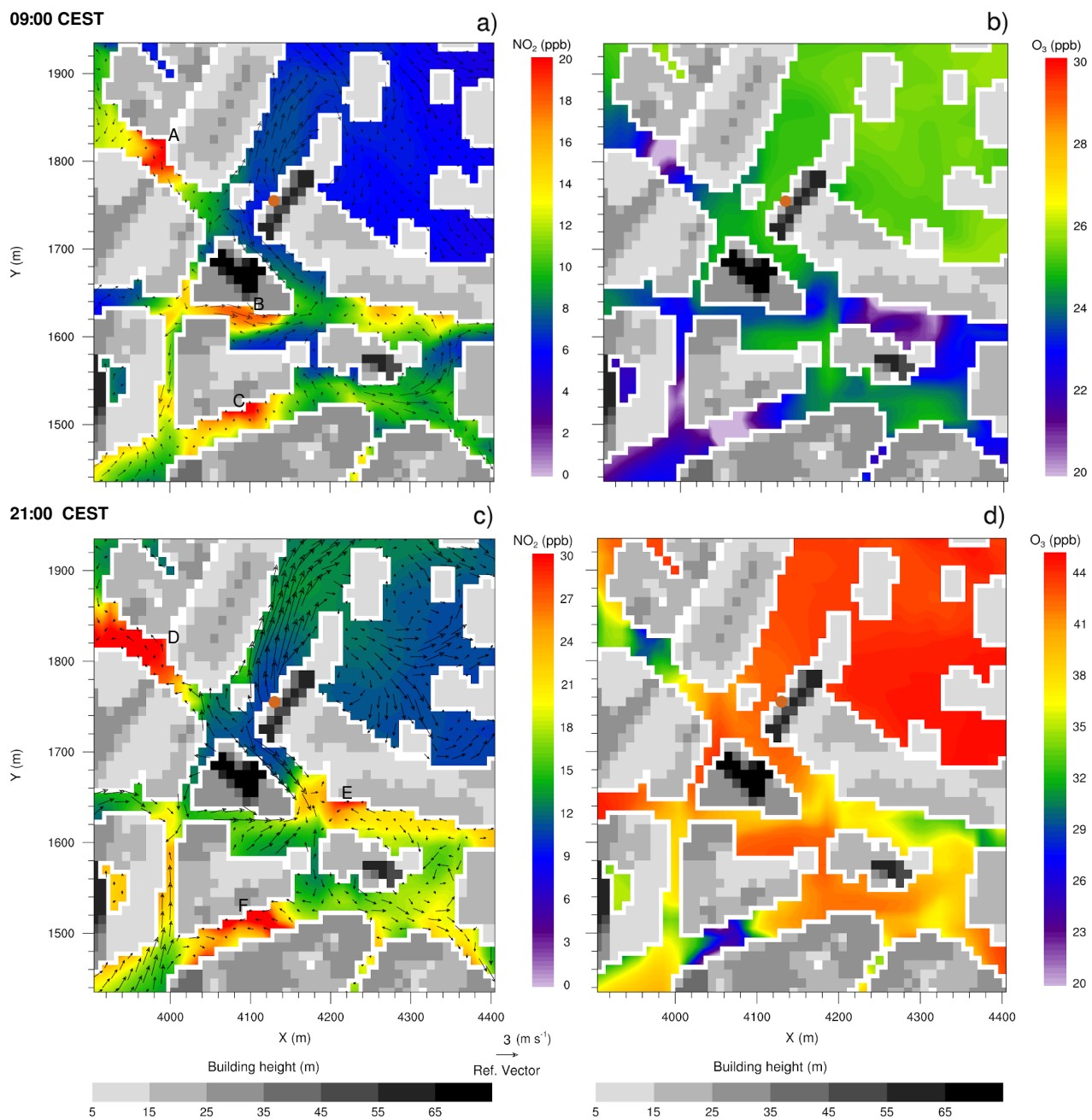

**Figure 7.** Near surface $NO_2$ and $O_3$ concentrations in the street canyons near Hardenbergplatz, simulated with the CBM4 mechanism at 09:00 and 21:00 hours on 17 July 2017. The wind vectors in "a)" and "c)" show horizontal wind speed 5 m above the surface. The brown filled dot, in "a)", indicates the location of Hardenbergplatz air quality station. Grey shading indicates building height.

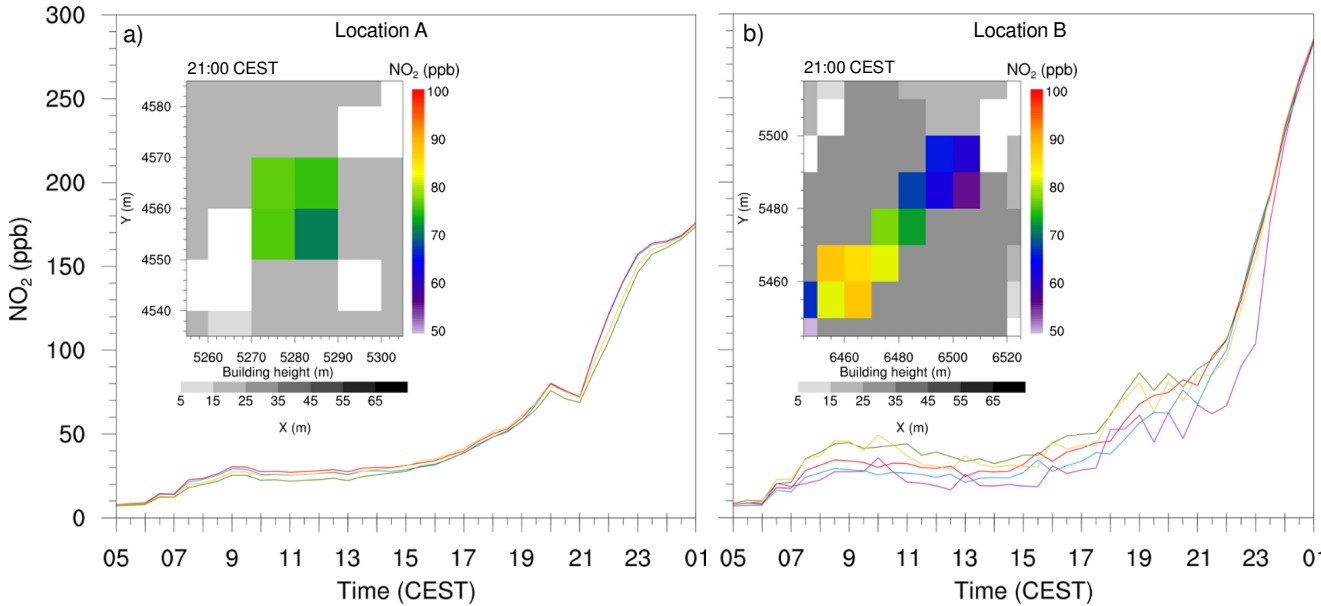

**Figure 8.** Time series plots from a) four individual grid points with high concentrations at location A, and b) five randomly selected grid points from the isolated street canyon at location B as indicated in Fig. 6c. The inset shows the exploded view of these isolated cavities at location A and B at 21:00 hours on 17 July 2017. Colours of the curves in the time series plots are randomly selected and has no relevance to the shading of the inset plots.

the evening hours when the surface layer stabilises. If, however, surface emissions are still high within such street canyons, chemical compounds can accumulate and will not be sufficiently transported away. Here, we emphasise that such accumulation of compounds cannot be observed with higher grid resolution of 1-2 m where street canyons are sufficiently resolved by the numerical grid.

### 4.4    Comparison of the three chemical mechanisms

Figure 9 shows the time-height plots of $NO_2$ and $O_3$ at Hardenbergplatz. All six plots exhibit a clear signal of gradual boundary layer growth from morning to evening. The concentrations of both $NO_2$ and $O_3$ differ between the three chemical mechanisms which can be attributed to their different levels of complexity and different description of the reactions resulting in either photochemical equilibrium or net formation of ozone and other photochemical products.

Due to the photostationary state, there is no net $O_3$ production in the PHSTAT mechanism (Fig. 9a and b). In the morning hours, despite reaction R1 that consumed most of the NO, the $NO_2$ concentration is slightly reduced to 7 ppb by 7:00 CEST compared to its initial near surface background levels of 10 ppb. The small drop in $NO_2$ concentration is attributed to the mixing of $NO_2$ within the shallow mixed layer reaching 500 m in depth. Titration of $O_3$ by NO, consequently reduces $O_3$ concentrations to 15 ppb. After 07:00 CEST, photodissociation leads to a decrease of $NO_2$ to very low levels (2-4 ppb) 50 m

above the surface. However, $NO_2$ concentration remains on the order of 6 to 7 ppb close to the surface due to emissions from roads and $NO_x$-VOC-$HO_x$ chemistry. Consequently, $O_3$ concentration peaks between 13:00 to 18:00 CEST by reaction R3. Above the rapidly growing mixed layer, the background concentration remained 2 and 40 ppb for $NO_2$ and $O_3$ respectively. In the evening and nighttime emissions from traffic, suppression of $NO_2$ photolysis (R2) and a slow R1 reaction increases $NO_2$ concentrations near the surface up to 20 ppb, while $O_3$ concentration is suppressed to less than 15 ppb due primarily to reaction R1. Above the stable nocturnal boundary layer the $NO_2$ is well mixed in the near neutral residual layer with a concentration of around 10 ppb, while due to absence of NO in the residual layer, and downward transport of background $O_3$ concentration through the entrainment zone, the $O_3$ concentration increases by 30 ppb in the residual layer.

Photolysis of $NO_2$, daytime $NO_x$-VOC chemistry in the case of the SMOG (Fig. 9c and d) and CBM4 (Fig. 9e and f) mechanisms result in $O_x$ production and increased $O_3$ levels during the day with maximum values in the afternoon. Various VOC reactions form peroxy radicals that oxidise NO to $NO_2$ and other nitrate products help increase $O_3$ concentration whereas CO, $CO_2$, and $HNO_3$ acts as atmospheric sink to aldehydes, $NO_x$ and $O_3$ in SMOG.

In the CBM4 mechanism $NO_2$ levels are reduced as much as 40 % between 12:00 to 19:00 CEST whereas $O_3$ concentration increases with almost the same percentage. Since no reactions leading to a net production of $O_3$ are considered for the photostationary state (PHSTAT mechanism), the maximum ozone concentration is determined by the photolysis of $NO_2$ and is therefore lower than for the SMOG and CBM4 mechanisms.

The interaction of $NO_x$ with $O_3$ chemistry is more pronounced near the surface. In all three mechanisms, a gradual decay of $O_3$ is found near the surface in the evening (after 20:00 CEST). During the daytime, specially and in particular, in the morning hours, entrainment of $O_3$ from the residual layer during the growth of the boundary layer contributes to the increase in $O_3$ concentration in the mixed layer. In the evening, after the collapse of the mixed layer, a well-mixed residual layer with near-uniform pollutant concentration can be found above the shallow, stably stratified nocturnal boundary layer. Although behaviour of $NO_x$ and $O_3$ is essentially the same in all three mechanisms, the difference is more evident in the CBM4 mechanism because we see a more pronounced diurnal course of $O_3$ in CBM4.

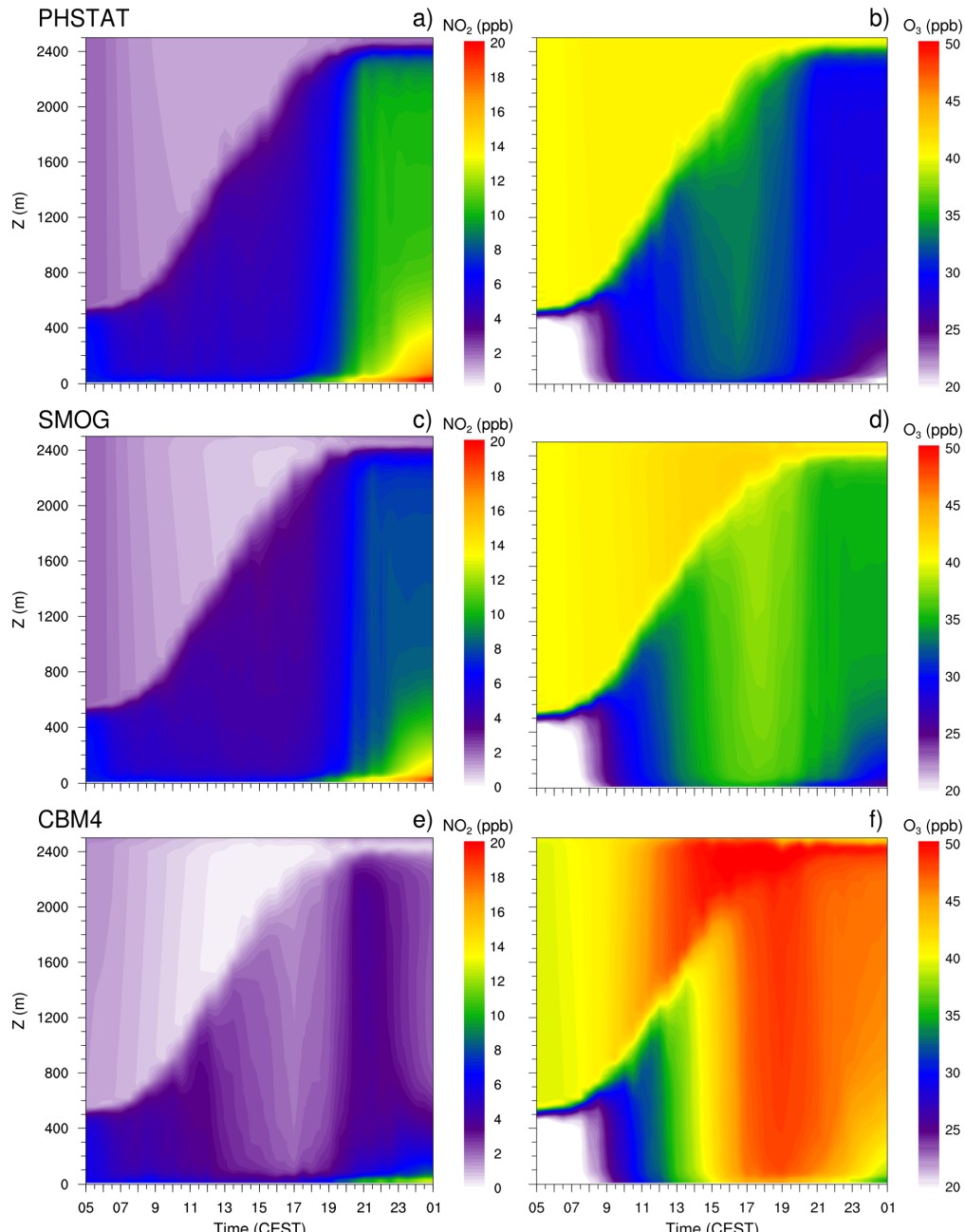

**Figure 9.** Time-height cross-section of half hourly averaged $NO_2$, and $O_3$ concentrations simulated with the PHSTAT, SMOG and CBM4 mechanisms from 5:00 CEST (17 July 2017) to 01:00 CEST (18 July 2017) at Hardenbergplatz. The left column, (panels a,c,e) shows $NO_2$ and the right column, (panel b,d,f) shows $O_3$ concentration.

## 4.5 Comparison of pollutant concentrations with observations

The observation data from Wedding and Hardenbergplatz is compared with the mean concentrations of the modelled NO, $NO_2$, and $O_3$ from five points in the vicinity of the Wedding and Hardenbergplatz stations (Fig. 10). Error bars indicate standard deviation at every hour in the time series data from the five points. Ozone is not measured at Hardenbergplatz, and therefore, the time series curve of the observed $O_3$ is not included in Fig. 10 and the analysis at Hardenbergplatz is restricted to NO and $NO_2$ species only.

The three mechanisms (PHSTAT, SMOG and CBM4) are able to reproduce the diurnal cycle of NO, $NO_2$, and $O_3$ with reasonable accuracy. The CBM4 mechanism shows a more pronounced diurnal course than SMOG and PHSTAT. The timing of the maximum ozone concentration is reproduced much better in the CBM4 mechanism than the SMOG and PHSTAT mechanisms. Since the PHSTAT mechanism does not include any net $O_3$ formation, the maximum $O_3$ concentration occurs at the time of maximum solar elevation. However, the SMOG and the CBM4 mechanisms include photochemical ozone production which leads to higher maximum ozone concentrations. A shift of the maximum ozone concentration from local noon to later afternoon is in agreement to observations. Due to a more detailed description of $NO_x$-VOC-$HO_x$ chemistry, this process is described more realistically for the CBM4 mechanism than for the SMOG mechanism. All three mechanisms failed to realistically simulate evening peaks in $NO_x$ (NO and $NO_2$) concentrations and the corresponding low $O_3$ concentrations. This can possibly be related to the too slow cooling of the surface as described earlier which results in less pronounced stable layer near the ground after sunset. A reason could be that the observed decrease of the wind speed to values below 1 m s$^{-1}$ between 22 and 24 h CEST could not be reproduced by the model due to the application of cyclic boundary conditions. Since $O_3$ is inversely related to $NO_x$, the underestimated $NO_x$-concentration in the evening leads to reduced titration of $O_3$, therefore, results in higher levels of $O_3$. Another reason for the bias of the model could be due to the uncertainties in the description of the traffic emissions. We utilized only parameterized traffic emissions, which may under-represent the emissions at the considered locations in the evening hours.

The no-reaction case only contains transport and dry deposition of $NO_x$. Without chemical reactions, NO continues to increase, which is primarily due to the lack of chemical sinks in combination with the application of cyclic boundary conditions. In contrast to the cases with chemical reactions, $NO_2$ is lower than NO since there is no conversion of NO to $NO_2$. Despite titration by NO, a very slow increase in the near surface $O_3$ in the morning hours can be attributed to the downward mixing from the residual layer during the growth of the mixed layer.

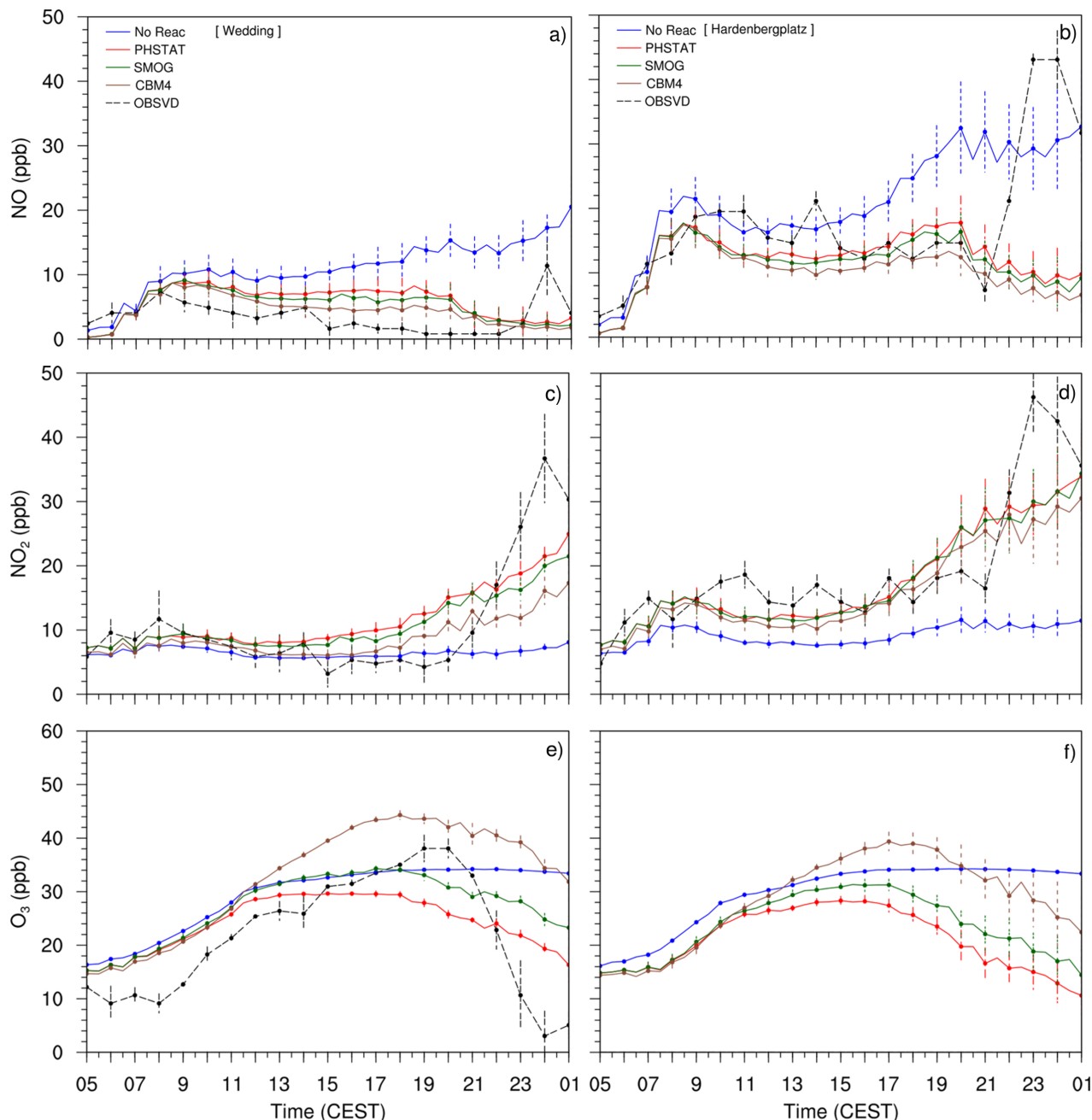

**Figure 10.** Time series plots of the near surface NO, NO$_2$, and O$_3$ concentrations from four mechanisms (NO-REAC, PHSTAT, SMOG, CBM4) and observations from 5:00 CEST (17 July) to 01:00 CEST (18 July 2017). The left column (panel plots a, c and e ) shows time series from Wedding, and the right column (plots b, d, and f) shows time series plots from Hardenbergplatz.

## 4.6 Computational efficiency

Chemistry models are known for utilizing a relatively large amount of computational power and the same applies to the chemistry model implemented in the PALM model. This is mainly due to the complexity in solving the stiff Jacobian matrix because reaction rates of chemical species vary greatly that leads to a time-step much smaller than meteorological parameters, therefore the computation speed of the entire system is largely affected. The simulation of the chemical reactions can take as much as 90 % of the total computational time in the calculation of an online-coupled simulation (Cao et al., 2019). The computation time of chemistry increases with increasing number of the chemical compounds and the number of chemical reactions in a given chemical mechanism. Table 2 shows a comparison of the increase in computational time for four chemical mechanisms relative to the corresponding meteorology only simulation on two different high performance computing systems and two different single domain extents.

For Case A, simulations were performed for a domain with 96 x 96 x 320 grid points and dx = dy = dz = 10 m. The model domain for the Case A simulations is a smaller part of the model domain shown in Figure 2 with the centre located at the Ernst-Reuter-Platz. The simulations for Case A are performed on the Karlsruhe Institute of Technology (IMK-IFU) cluster on 96 CPU's of the intel Ivy-bridge processors. Case B consists of the simulations which are discussed in the previous sections. The configuration and domain size is described in section 3.1. These simulations were performed at the North-German Super computing Alliance (HLRN)-Berlin with 784 CPUs on Intel Cascade Lake Platinum 9242 CPUs (CLX-AP) on standard96 partition. Both cases have been integrated for 24 hours. Due to the limited computational resources, the 'transp. only' option (i.e. no-reaction; species used as passive scalars only) for all mechanisms was applied only for case A. As for 'transp. only' simulations for case B, only the single run for the PHSTAT mechanism from section 4.5 was available.

The results show a significant increase in the computational cost relative to the meteorology only simulation for the same model domain. The comparison of 'transp. only' and '(full)' for Case A ( Table 2) shows that the transport of additional scalar variables is even more expensive compared to the computation of the chemical transformation. While the increase in computational costs for the transport increases linearly with the number of compounds, this is not the case for computation of chemical conversion. As described by Verwer et al. (1999) the efficiency of the applied Rosenbrock solver increases with increasing size of the chemical mechanism. Comparison of 'transp. only' of case B with 'full' suggests that for the PHSTAT mechanism the computational expense for the transport of the additional scalars is almost the same as for the computation of the chemical transformation. Although the 'transp. only' simulation for case B was only performed for PHSTAT, presumably the relative computational expense for 'transp. only' and 'full' for the other mechanisms would not be much different for case A and case B.

Given the highest number of chemical compounds and consequently the number of reactions, the CBM4 mechanism is the most resource-intensive of all mechanisms, which is 5.5 times more expensive than a plain meteorology-only simulation for the small domain (case A), and 6.5 times more expensive for the large domain simulation (case B). Considering the high computational resource requirement of chemistry-coupled LES simulations, it is important to invest extra efforts in the design of the simulation and selection of the appropriate chemical mechanisms.

**Table 2.** Increase in computational expense of a simulation with the PASSIVE, PHSTAT, SMOG and CBM4 mechanisms relative to a simulation without chemistry. For Case A, the simulations for a model domain with 96 x 96 x 320 grid points and a grid width of 10 m. For Case B, the simulations were performed for a model domain with 671 x 671 x 312 grid points and a grid width of 10 m. Both runs were made in scalar mode for one full day. 'transp. only' indicates the relative expense when solely the transport of the considered compounds is calculated and chemical transformation is switched off, 'full' refers to the run with transport as well as chemical transformations.

| Mechanism | Var. species | Reactions | Increase in computational time (%) | | | |
| --- | --- | --- | --- | --- | --- | --- |
| | | | Case A (transp. only) | Case B (transp. only) | Case A (full) | Case B (full) |
| PASSIVE | 2 | 0 | 20 | – | – | – |
| PHSTAT | 3 | 2 | 30 | 28 | 50 | 63 |
| SMOG | 13 | 12 | 120 | – | 190 | 310 |
| CBM4 | 32 | 81 | 310 | – | 550 | 650 |

## 5 Concluding remarks

We have outlined the structure and important features of a chemistry model that has been added to the PALM model system 6.0 as part of PALM-4U (PALM for urban applications) components and coupled to the PALM model core. PALM model system 6.0, is the first LES model that can simulate chemical transformation, advection and deposition of air pollutants for larger and realistically shaped urban areas (Maronga et al., 2019). PALM 6.0, offers many advanced features, which were not included in other LES models, for example self-nesting and large-scale forcing for chemical species, flexibility of choice in selection and generation of gas-phase chemistry mechanisms with the KPP preprocessor, and the availability of the chemistry model in both, RANS and LES mode.

The results of the presented case study demonstrate the ability of the chemistry model to predict photochemical reactions, along with advection and deposition of pollutants. The difference between reactive and non-reactive case clearly indicates that invoking reactive chemistry is of critical importance to accurately predict pollutant concentrations at the microscale. The simulated concentrations follow the observed diurnal cycle of the pollutants. However, the weak agreement warrants an improvement in the description of the parameterized emissions, the large-scale forcing and model resolution.

Especially with coarse grid resolution some street canyons are only poorly resolved by the numerical grid, occasionally leading to the situation that street canyons are only one grid point wide or isolated cavities occur. However, within such enclosed cavities the flow is only poorly resolved too, meaning that the total transport of energy and matter is only small, especially under stable conditions. Emissions at such enclosed geometries may result in unrealistically high concentrations of primary compounds like $NO_x$. With increasing grid resolution such narrow street canyons will be better represented on the numerical grid, lowering the risk that high emissions coincide with regions where the flow is only poorly resolved. Hence, especially for lower grid resolutions, we recommend to carefully inspect the concentration output to treat high concentrations in the data analysis adequately.

The computational cost of the chemistry coupled simulations largely depends on the number of chemical compounds and the number of chemical reactions. The transport of additional scalar variables are found more expensive than the computation of the chemical transformation. Considering the high computational demand of chemistry coupled LES simulations, it is important that users should be careful in the choice of the chemistry mechanism for their specific needs.

Although the maximum ozone concentration and the time of its occurrence is somewhat better reproduced with the CBM4 mechanism than with the SMOG mechanism, it is difficult to give a final recommendation for an optimum mechanism. For some applications – in particular for low VOC conditions – SMOG or even the photostationary equilibrium may be sufficient while for other simulations the application of the CBM4 mechanism is simply not possible due to its high computational demand. However, when using strongly condensed mechanisms like SMOG with only one single VOC compound, the user

must always keep in mind that the simulated concentrations of ozone and further oxidation products depend strongly on the rate constant for the reaction of this single lumped VOC with OH. As already elaborated by Middleton et al. (1990), the reactivity of the lumped VOC depends on the VOC mix and therefore on the local emissions. On the other hand, CBM4 is still too simple for investigations with special focus on VOC chemistry or a prediction of semi- and non-volatile organics, which are required for the SALSA aerosol model. In this case more detailed or advanced chemistry mechanisms than CBM4 may be required,

which can be easily added since the flexible configuration of the chemistry in the PALM model allows the easy implementation of new mechanisms.

The chemistry model implemented in PALM is under continuous development, new features are being added and current features are being improved both in terms of efficiency and accuracy. We plan to further extend the model chemistry by implementing an advanced biogenic volatile organic compounds (BVOC) emission scheme, emission parameterization of

residential heating, biogenic pollen emission, transport and deposition scheme. Moreover, the chemistry model will be further improved by including a simplified scheme for ultra-fine particles (UFP) besides the one already implemented in SALSA (Kurppa et al., 2019). Additionally, chemical boundary conditions, road-re-suspension processes, wet-deposition processes of air pollutants and a photolysis parameterization based on shading information will be implemented in the chemistry model. We also plan to provide a more complex and detailed chemistry for the RANS mode, while for the LES mode, currently only

strongly simplified mechanisms are possible due to computational constraints.

*Code and data availability*. The PALM source code including PALM-4U components is freely available under the GNU General Public License v3. The exact model source code of the two revisions (4450 and 4601) is available at https://palm.muk.uni-hannover.de/trac/browser?rev=4450 (last access: 9 Sept 2020) and https://palm.muk.uni-hannover.de/trac/browser?rev=4601. The documentation for the chemistry model is available at https://palm.muk.uni-hannover.de/trac/wiki/doc/app/chemdesc. The input data required to

run chemistry coupled PALM simulations used in this study and the observed pollutant data used for comparison are freely available at https://doi.org/10.5281/zenodo.4153388 (Khan, 2020).

*Supplement*: The supplement related to this article is available online at: https://doi.org/10.5194/gmd-2020-286-supplement

*Author contributions.* All authors contributed to the development of the presented Atmospheric Chemistry Model. All authors contributed to the text and the presented analysis. BK created the general structure of this article and conducted the presented simulations.

*Competing interests.* The authors declare that they have no conflict of interest.

*Acknowledgements.* MOSAIK and MOSAIK-2 are funded by the German Federal Ministry of Education and Research (BMBF) under grant 01LP1601 and 01LP1911H within the framework of Research for Sustainable Development (FONA; www.fona.de). This work was supported by the North-German Supercomputing Alliance (HLRN). We are grateful to the HLRN supercomputer staff, especially Dr.-Ing. Stefan Wollny for his continual help and support. Wieke Heldens and Julian Zeidler of the German Aerospace Center (DLR) provided the Geographical information data, Kristina Winderlich and Eckhard Kadasch of German Meteorological Service (DWD) provided preprocessed COSMO-DE data and developed INIFOR software utility to make COSMO data readable to PALM

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
