# Peer review of "Development of an atmospheric chemistry model coupled to the PALM model system 6.0: Implementation and first applications"

_Geoscientific Model Development, 2020_

## Short Comment (SC1) · 27 Oct 2020

Dear authors,

in my role as Executive editor of GMD, I would like to bring to your attention our Editorial version 1.2:

https://www.geosci-model-dev.net/12/2215/2019/

This highlights some requirements of papers published in GMD, which is also available on the GMD website in the 'Manuscript Types' section: http://www.geoscientific-model-development.net/submission/manuscript_types.html

In particular, please note that for your paper, the following requirement has not been met in the Discussions paper:

- Code must be published on a persistent public archive with a unique identifier for the exact model version described in the paper or uploaded to the supplement, unless this is impossible for reasons beyond the control of authors. All papers must include a section, at the end of the paper, entitled "Code availability". Here, either instructions for obtaining the code, or the reasons why the code is not available should be clearly stated. It is preferred for the code to be uploaded as a supplement or to be made available at a data repository with an associated DOI (digital object identifier) for the exact model version described in the paper. Alternatively, for established models, there may be an existing means of accessing the code through a particular system. In this case, there must exist a means of permanently accessing the precise model version described in the paper. In some cases, authors may prefer to put models on their own website, or to act as a point of contact for obtaining the code. Given the impermanence of websites and email addresses, this is not encouraged, and authors should consider improving the availability with a more permanent arrangement. Making code available through personal websites or via email contact to the authors is not sufficient. After the paper is accepted the model archive should be updated to include a link to the GMD paper.

Note, that revision tags of an institutionally hosted website are not permanent archives. Please upload the 2 version refering to the revision tags in persistent archives providing a DOI or other permanent identifier (e.g. in zenodo).

Yours, Astrid Kerkweg

―――――――――――――――――

---

## Short Comment (SC2) · 29 Oct 2020

Dear Basit Khan,

thank you for generating DOIs for your source code!

As you posted the respective URL in the response, I think it is sufficient to just change the link in the manuscript upon submission of the revised version.

Best regards,

Astrid Kerkweg

---

## Author Comment (AC1) · 29 Oct 2020

Dear Astrid Kerkweg,

Thanks for the comment. I have uploaded revision 4450 and 4601 of the PALM model source code on Zenodo. However, due to addition/upload of new files, I had to create the new version and therefore, a new DOI is issued for this updated version of the published data. The url of the new version is https://doi.org/10.5281/zenodo.4153388.

I wonder if I need to send you the revised manuscript with the new DOI? Please advise.

Thanks, stay safe and best wishes,

[Figure]

Basit Khan and Co-authors.

---

## Referee Comment (RC1) · Anonymous Referee #1 · 11 Nov 2020

Recommendation: Minor revision

General comments:

The current manuscript describes a new city-scale LES-Chem model, PALM-4U and presents test simulation results with a variety of chemistry modules. It is indeed important to have a variety of chemistry modules from simple to complex in one model, so that each user can allocate limited computational resources to his/her own specific targets. Some might need higher resolutions or longer time integrations, while some might need very accurate chemical modules for coarser resolutions or shorter time integrations. The quality of their work meets the standard of GMDD but there are several

issues remained to improve the presentation of manuscript as listed in the following specific comments.

Specific comments:

(1) The country names are missing in the affiliations of co-authors from #2 to #6.

(2) There are several abbreviations in abstract without being defined, such as PALM, PARAMETERIZED, CBM4, SMOG, and PHSTAT. PALM is their model name but it is never spelled out throughout the manuscript.

(3) There are more abbreviations in the entire manuscript, such as MITRAS, ASMUS, SALSA, etc. Better to spell them out when they appeared first time, or make a table of nomenclature.

(4) The relationship or difference between PALM and PALM-4U is unclear in the entire manuscript. Sometimes the author write PALM, but sometimes PALM-4U. It can be read that PALM consists of PALM-4U and the chemistry module was in PALM-4U, not PALM (Sect.2.1 and Fig. 1). However, in Sect. 2.5.5, it seems that the chemistry module was with PALM, and PALM-4U was not appeared. In Sect. 2.5.6 as well. It is a bit confusing. Please clarify the relationship or difference between the two models and be accurate in the definition throughout the manuscript.

(5) P.4 Ln. 16: "Optical cloud and rain water" may require explanation. It was written later that cloud microphysics was not implemented, so what are they?

(6) Figure 1: Arrows are ambiguous. What's the difference between grey arrow and green arrows? What's the difference between the green arrow with single direction and bi-direction? It seems that emission to chemistry driver is one-way, but there is a direction from chemistry toward emission. What process is it?

(7) P.6, L.15 "deposition, scavenging" -> "deposition and scavenging"

(8) P.8, L.13 "Fast-J" may need reference. If it is an abbreviation, please spell it out.

(9) P.8, L.20-21; for aerosol phase, better to write sulfate (SO42-), nitrate (NO3-), and ammonium (NH4+).

(10) P.9, L.10: "following (Simpson et al., 2003)." -> "following Simpson et al. (2013)."

(11) P.9, L.29: "The chemistry model of PALM" -> "The chemistry model of PALM-4U", right?

(12) P. 10, L.12 "modes": "sectors" are more frequently used. Please consider to rephrase.

(13) Sect. 3.1 "numerical set-up" includes several sentences which should be described in different sections.

a) P. 11, L. 2: "Details of the dynamics core ... Maronga et al. (2020)" better to be moved to a model description section, Sect. 2.x.

b) 2nd paragraph of Sect. 3.1, the first two sentences "Observations from ... a 24 hour run." and "The ceilometer observations ... the diurnal cycle" and the latter two sentences "Fig. 2 shows ..." and "The horizontal grid spacing..." are not relating with each other. The description of weather by Ceilometer observation was already mentioned in the 1st paragraph of Sect. 3.1. Better to reorganize the 1st and 2nd paragraph of Sect. 3.1.

c) P. 11, L. 26, "A third order Runge-Kutta, ... (Wicker and Skamarock, 2002)" are already written previously.

(14) P. 11, L.12: Spell "TU" out here.

(15) P.11, L.165: "Ceilometere" -> "Ceilometer"

(16) P. 12, L. 2: Probably "(Resler et al., 2017; Maronga et al. 2020)" looks better.

(17) P. 12, L.16: "Monin-Obukhov Similarity Theory (MOST)" mentioned already several times in the previous locations. Define MOST when it is first appeared and use

MOST in the following locations.

(18) P.13, L.25-35: How the authors set the boundary conditions for chemical species are not explained.

(19) Sect. 3.2: Please provide the heights of the observation sites, Wedding and Hardenbergplatz. In urban locations, the observation points could be on the roof of building. If this is the case, comparison between the simulated 5-m height concentration and the observed roof-top concentration are inconsistent.

(20) Sect. 4.2: The chemistry module and the grid point used to depict Fig. 4 was missing. Probably CBM4 and Hardenbergplatz, though.

(21) P. 24, L. 5: "Figure 10" -> "Fig. 10"

(22) Table 2: It is quite reasonable that the CPU time of transport only without meteorology of CBM4 was 10 times that of PHSTAT (310/30) because the number of tracers is also 10 times (32/3). However, with chemical reactions, why the CPU time of CBM4 was still 10 times that of PHSTAT (550/50) even though the number of chemical reactions of CBM4 (81) is 40 times that of PHSTAT (2). Is it because only two reactions with KPP requires as much CPU time as 81 reactions?

(23) Overall, the authors show horizontal variations in concentrations in Figs. 8 and 10 and vertical profiles from 0 m to 2,500 m in Figs. 3, 4, 5, and 9. However, they did not show the vertical profiles in the bottom layers (i.e., below 50 – 100 m), even though there seems very sharp vertical gradients. This might be of interest. Is it possible for the authors to show the horizontal distributions of concentrations near the top of urban canopy (at several ten meters?) to compare and discuss the differences from those at 5 m of Fig. 6, for example? Large scale models can only simulate the concentration above the urban canopy and many of the urban observation points exist on the roof of buildings. It is very informative for large scale modelers to show the difference in concentrations between the street canyon and urban canopy top.

---

## Referee Comment (RC2) · Anonymous Referee #2 · 16 Nov 2020

The paper describes the new PALM6.0 model system, an extension of the existing PALM model with on-line gas-phase chemistry (several schemes, with different levels of complexity) and deposition. The functioning of the system is evaluated for the different chemical schemes for test cases in Berlin. In addition the computational costs of the different levels of complexity are analysed. The paper is quite well-structured and complete. Case studies are interpreted in detail and compared to observations. Figures are relevant and to the point. All relevant aspects are described. However, at some places further clarification is needed. Also the English needs improvement. All in all I compliment the authors with their nice work and I'm looking forward to the final version.

Detailed comments

Abstract Since you name the chemical mechanisms at the end of the abstract you dan also directly name them in the beginning of the abstract and rank them in complexity. Naming of emission mode is not so relevant here. The authors state that this is the first paper with complex gas phase chemistry at this high resolution in an on-line coupled model for an urban geometry. However, the feedback of modelled concentrations on meteorology is not elaborated here. The authors could say a few words on this.

P3: comment: Li et al 2016 also included rather detailed chemistry, but indeed more simple than CBM-IV

P7: CBM-IV is replaced by more detailed CB5 and CB6 mechanisms, so not that widely used any more, but you could give the valid argument that it is the lightest version of 'full' gas-phase chemistry.

P8 l20 confusing sentence, compounds are not all in the particulate phase

P11 Numerical set-up description. The authors switching between description of case, used observations and numerical set-up makes it more difficult to follow or look things up.

P11 L8 It was not a weekend, therefore, emissions from the traffic were not affected by the reduced traffic Which reduced traffic? You mean that the day is a working day with the corresponding traffic activity, which is higher than in a weekend. The topic is addressed better on p 11 and p13. This sentence is confusing

P 12 L3-4 in one sentence 50 and 60 m used, why this criterium? What is a few?

P13 Observed NO, NO2 and O3 from Berlin city, are these the stations in section 3.2? Did you interpolate or take one value for the domain?

P15: This part about chemistry needs some reformulation and better explanations. R3 should also contain an M on the left side.P16 l6 The authors state that NO and NO2 do

not lead to a net gain in O3. But that is not completely true, the final photostationary equilibrium depends on NO2/NO ratio and changing emissions/concentration leads to a change in O3, as correctly stated on p18l12. VOC plays an additional role. For an urban area with NOx abundance, VOC is mostly the limiting factor for O3 formation. Also in R4, the meaning of RH and RO2 are not explained in the text. The additional impact of VOC is indeed visible in Figure 10.

P17, section 4.3 Spatial distribution of pollutants: why do you switch to CBM4 here, whereas SMOG was the chosen to be the default chemical scheme? 4.2 was with SMOG (as I understood from the context and the model description: SMOG was chosen. . .p13 l8)

P22 l 28 Downwelling in the entrainment zone: I would call this entrainment, mixing in of air when the boundary layer rises. See also p24 last sentence.

P24 l 18 Missing emission sources: would you expect that the contributions from industry, household and BVOC would have a significant peak in the evening in summer? I doubt this.

P25 Section 4.6 Numerical efficiency test, are case A and B related to a specific location? Is domain A included in B, for which day?

P27 Concluding remarks. Now that the chemistry schemes are compared in terms of performance with respect to observations and computational effort, could you conclude whether SMOG or CBM-IV would be the best default option? Especially in the light of the further complexity of the model (BVOC, SALSA) and the last sentence. How does your conclusion relate to practices in other LES models?

Typo's/text corrections P10 l6 sentence with mostly sunny with. . .

P10: Meteo from web sources: this is meteo from the airport published at a website I assume. When I read the sentence I'm in doubt of the data source and quality.

L 16 Ceilometere

P11, p12 Some brackets missing in references

P12 L9-11 confusing and..and.. with respectively, unclear what is meant exactly

P14 output-> put out/written to file. Output was exported to file every 10 minutes. . ..

P22 l 31-32 Although. . .., however. . . Use one of the two

---

## Author Comment (AC2) · 4 Dec 2020

[11pt, a4paper]article [a4paper, left=3.5cm, right=2.5cm, top=2.0cm, bottom=2.5cm]geometry amssymb,amsmath [utf8]inputenc [T1]fontenc enumerate fancyhdr graphicx mathptmx units [authoryear]natbib hyperref pgfgantt microtype

**Response to comments of the anonymous Referee #1, "RC1"**

*Comment (1): The current manuscript describes a new city-scale LES-Chem model, PALM-4U and presents test simulation results with a variety of chemistry modules. It is indeed important to have a variety of chemistry modules from simple to complex in one model, so that each user can allocate limited computational resources to his/her own specific targets. Some might need higher resolutions or longer time integrations, while some might need very accurate chemical modules for coarser resolutions or shorter time integrations. The quality of their work meets the standard of GMDD but there are several issues remained to improve the presentation of manuscript as listed in the following specific comments.*

**Author's resp:** First of all, we would like to thank the reviewer for the effort in reviewing our paper and the valuable detailed comments, which helped us to improve the paper. Please note that page and line numbers mentioned in responses to comments correspond to the attached updated version of the manuscript

*Comment (1) The country names are missing in the affiliations of co-authors from #2 to #6.*
**Author's resp:** Country names have been added to the affiliation text.

*Comment(2) There are several abbreviations in abstract without being defined, such as PALM, PARAMETERIZED, CBM4, SMOG, and PHSTAT. PALM is their model name but it is never spelled out throughout the manuscript.*

**Author's resp:** PALM, and chemistry mechanisms have now been defined when the first time they appear in the text. Emission mode 'PARAMETERIZED' is defined briefly in the abstract and then in greater detail in section 2.3 (Pg 1, Ln 2, Ln 8-10, Pg 10, Ln 1-2).

"...PALM model system 6.0 (formerly an abbreviation for Parallelized Large-eddy Simulation Model and now an independent name)... "

"... ranging from the photostationary state (PHSTAT) to mechanisms with a strongly simplified VOC chemistry(e.g. the SMOG mechanism from KPP) and the Carbon bond mechanism (CBM4, Gery et al. (1989b)), which includes major pollutants namely O3, NO, NO2, CO, a more comprehensive, but still simplified VOC chemistry and several products ... "

*Comment(3) There are more abbreviations in the entire manuscript, such as MITRAS, ASMUS, SALSA, etc. Better to spell them out when they appeared first time, or make a table of nomenclature.*

**Author's resp:** Abbreviations have been spelled out in the text (Pg 2, Ln 31-32; Pg 4, Ln 9).

"...and trees e.g. MITRAS (the microscale obstacle resolving transport and stream model), (Salim et al., 2018) and ASMUS (A numerical model for simulations of wind and pollutant dispersion around individual buildings), (Gross, 1997),..."

"... SALSA (A sectional aerosol module for large scale applications), (Kokkola et al., 2008) implemented in PALM ..."

*Comment(4) The relationship or difference between PALM and PALM-4U is unclear in*

*the entire manuscript. Sometimes the author write PALM, but sometimes PALM-4U. It can be read that PALM consists of PALM-4U and the chemistry module was in PALM-4U, not PALM (Sect.2.1 and Fig. 1). However, in Sect. 2.5.5, it seems that the chemistry module was with PALM, and PALM-4U was not appeared. In Sect. 2.5.6 as well. It is a bit confusing. Please clarify the relationship or difference between the two models and be accurate in the definition throughout the manuscript.*

**Author's resp:** PALM is primarily an LES code with an additional RANS mode. The PALM model comprised of a dynamic core and various embedded models. PALM-4U is the conceptual framework of the model components embedded/coupled to PALM which are specifically designed and/or used for urban applications. The description of PALM and PALM-4U has been rephrased in the text (abstract and in section 2.1) so that to make the difference more clear and easy to understand (Pg 1, Ln 4-5; Pg 4, Ln 21-23).

"... The new chemistry model is implemented in the PALM model as part of the PALM-4U components (read: PALM for you; PALM for urban applications) which are designed for application of PALM model in the urban environment (Maronga et al., 2020)..."

Subsection 2.5.5 and 2.5.6 does not exist in the manuscript, therefore, the issue raised by the reviewer cannot be addressed.

*Comment(5) P.4 Ln. 16: "Optical cloud and rain water" may require explanation. It was written later that cloud microphysics was not implemented, so what are they?*

**Author's resp:** We never wrote that cloud microphysics was not implemented. Since PALM version 4.0 a bulk liquid-phase two-moment microphysics scheme is available which predicts rain droplet number concentration and rain water mixing ratio. To avoid any confusion, the text has been slightly rephrased (Pg 4, Ln 24-26).

" ... the three velocity components ($u$, $v$, $w$) on staggered Arakawa C grid, and four
scalar variables namely potential temperature ($\theta$) , water vapour mixing ratio ($qv$), a passive scalar s and the subgrid-scale turbulent kinetic energy (SGS-TKE) $e$ (in LES mode) (Maronga et al., 2019, 2020)..."

*Comment(6) Figure 1: Arrows are ambiguous. What's the difference between grey arrow and green arrows? What's the difference between the green arrow with single direction and bi-direction? It seems that emission to chemistry driver is one-way, but there is a direction from chemistry toward emission. What process is it?*
**Author's resp:** Green and thick grey arrows were added to improve presentation of the flow chart. However, to avoid any confusion, all arrows are now drawn in grey colour. The thin arrows show the general flow and direction of the data within the Chemistry model and its components whereas thick grey arrow indicates coupling of the Chemistry driver with the PALM model core (Pg 5). Figure 1 is uploaded for review.

*Comment(7) P.6, L.15 "deposition, scavenging" -> "deposition and scavenging"*
**Author's resp:** The text is updated (Pg 6, Ln 15).
".... (i.e. deposition and scavenging)."

*Comment(8) P.8, L.13 "Fast-J" may need reference. If it is an abbreviation, please spell it out.*
**Author's resp:** The text is updated and an appropriate reference Wild et al. (2000), has been added (Pg 8, Ln 5-6).
"... More extensive photolysis schemes such as the Fast-J photolysis scheme (Wild et al., 2000) that are based on the radiative transfer modelling will be included in the future..."

*Comment(9) P.8, L.20-21; for aerosol phase, better to write sulfate (SO42-), nitrate*

*(NO3-), and ammonium (NH4+).*
**Author's resp:** The text has been updated (Pg 8, Ln 13-14).
"... following chemical compounds in the particulate phase: sulphate ($SO_4^{2+}$), organic carbon (OC), black carbon (BC), nitrate ($NO_3^-$), ammonium ($NH_4^+$), sea salt, dust and water ($H_2O$) ..."

*Comment(10) P.9, L.10: "following (Simpson et al., 2003)." -> "following Simpson et al. (2013)."*
**Author's resp:** The reference has been updated (Pg 9, Ln 12).
" ... calculated following Simpson et al. (2003)... "

*Comment(11) P.9, L.29: "The chemistry model of PALM" -> "The chemistry model of PALM-4U", right?*
**Author's resp:** Yes, this is right. the text has been updated (Pg 9, Ln 30).
"... The chemistry model of PALM-4U includes a module for reading ..."

*Comment(12) P. 10, L.12 "modes": "sectors" are more frequently used. Please consider to rephrase.*
**Author's resp:** Thanks to point this out. We agree this sentence was slightly misleading. We have rephrased and improved the text (Pg 10, Ln 11-13 ).
"... More detailed traffic emission data can be provided in gridded form in PALM-specific NetCDF files (Maronga et al., 2020). LOD 1 emissions are gridded annual emission data for each sector (e.g. industry, domestic heating, traffic), which will be temporally disaggregated using sector-specific standard time factors..."

*Comment(13) Sect. 3.1 "numerical set-up" includes several sentences which should be described in different sections.*

*a) P. 11, L. 2: "Details of the dynamics core ... Maronga et al. (2020)" better to be moved to a model description section, Sect. 2.x.*

*b) 2nd paragraph of Sect. 3.1, the first two sentences "Observations from ... a 24 hour run." and "The ceilometer observations ... the diurnal cycle" and the latter two sentences "Fig. 2 shows ..." and "The horizontal grid spacing..." are not relating with each other. The description of weather by Ceilometer observation was already mentioned in the 1st paragraph of Sect. 3.1. Better to reorganise the 1st and 2nd paragraph of Sect. 3.1.*

*c) P. 11, L. 26, "A third order Runge-Kutta, ... (Wicker and Skamarock, 2002)" are already written previously.*

**Author's resp:** a) "Details of the dynamic core …..", has been moved to model description section 2.1 (Pg 4, Ln 32-33).

b) We have re-organised this section completely. Subsection 3.1 also renamed as " Modelled episode and modelling domain". Two further subsections have been introduced in order to improve the readability of this section (Pg 11-14 Ln – ).

c) This is probably fine. The first time Runge-Kutta and Wicker and Skamarock appeared in the general model description while on the second time, both appeared specifically as part of the model setup for simulations.

*Comment(14) P. 11, L.12: Spell "TU" out here.*

**Author's resp:** The abbreviation (TU) is removed and the actual name "Technical University" has been used instead (Pg 12, Ln 13). "... Charlottenburg building of the Technical University of Berlin ..."

*Comment(15) P.11, L.165: "Ceilometere" -> "Ceilometer"*

**Author's resp:** The spelling has been corrected (Pg 12, Ln 18).

*Comment(16) P. 12, L. 2: Probably "(Resler et al., 2017; Maronga et al. 2020)" looks*

*better.*
**Author's resp:** Both of the citations have been updated in the text (Pg 13, Ln 18).
"... the urban-surface model (Resler et al., 2017; Maronga et al., 2020)."

*Comment(17) P. 12, L.16: "Monin-Obukhov Similarity Theory (MOST)" mentioned already several times in the previous locations. Define MOST when it is first appeared and use MOST in the following locations.*
**Author's resp:** The Monin-obukhov similarity theory is abbreviated as MOST, the first time it appeared in the text, and then only "MOST" used in the following text (Pg 4, Ln 28).

*Comment(18) P.13, L.25-35: How the authors set the boundary conditions for chemical species are not explained.*
**Author's resp:** The description of the chemical boundary conditions on lateral, top and bottom boundaries has been further improved to make it more clear and easy to understand (Pg 14 Ln 19-20; Pg 14, Ln 29-31).
"... At the bottom boundary, a Dirichlet condition is applied to flow, $\theta$, and $q$ whereas a Neumann condition is applied to $e$, $p$ and chemical compounds. Moreover, a canopy drag coefficient $C_d$ = 0.3 has been applied while the roughness is specified internally depending on vegetation type. At the top boundary, Dirichlet boundary conditions are applied to flow and and $p$ only, initial gradient is applied to $\theta$ while Neumann boundary conditions are applied to $q$ and chemical compounds..."

*Comment(19) Sect. 3.2: Please provide the heights of the observation sites, Wedding and Hardenbergplatz. In urban locations, the observation points could be on the roof of building. If this is the case, comparison between the simulated 5-m height concentration and the observed roof-top concentration are inconsistent.*
**Author's resp:** The average height of air quality sensors is around 4 m above ground.

Therefore, comparison between observations and model data which is extracted 5 m above ground is consistent. The text has been updated in section 3.2 (Pg 12, Ln 26). "... average height of the air quality sensors at both stations is 4 m above ground..."

*Comment(20) Sect. 4.2: The chemistry module and the grid point used to depict Fig. 4 was missing. Probably CBM4 and Hardenbergplatz, though.*
**Author's resp:**: These are the spatial mean profiles of concentration and fluxes. The text and figure caption has been updated by adding "mean" and "simulated with CBM4 mechanism" (Pg 15, Ln 17; Pg 16, Ln–).
"Figure 3.Vertical profiles of a) potential temperature, b) mixing ratio, c) wind speed and d) wind direction, at different times from morning to midnight on 17 July 2017. The horizontal bars in a) indicate the boundary-layer height derived from ceilometer observations."
"... Figure 4 shows mean profiles of concentrations and vertical fluxes of NO, NO2, O3and CO for the selected times of the diurnal cycle on 17 July 2017, simulated with the CBM4 mechanism..."

*Comment(21) P. 24, L. 5: "Figure 10" -> "Fig. 10"*
**Author's resp:** The inconsistency in the use of '$Figure'$', and '$Fig.'$', has been removed. In the beginning of the sentence we use '$Figure'$', whereas within the sentence we use '$Fig.'$'. The use of both is now consistent throughout the manuscript.

*Comment(22) Table 2: It is quite reasonable that the CPU time of transport only without meteorology of CBM4 was 10 times that of PHSTAT (310/30) because the number of tracers is also 10 times (32/3). However, with chemical reactions, why the CPU time of CBM4 was still 10 times that of PHSTAT (550/50) even though the number of chemical reactions of CBM4 (81) is 40 times that of PHSTAT (2). Is it because only two reactions with KPP requires as much CPU time as 81 reactions?*

**Author's resp:** It is correct that CBM4 is 10 times more expensive than PHSTAT both for 'transport-only' and for 'Full' (transport + reactions) simulations. The relative increase of the computational expense for the solution of the chemistry equations does indeed decrease with increasing number of chemical compounds. As explained in section 3.4 of Verwer et al. (1999), the efficiency of the applied Rosenbrock solver increases with increasing size of the chemical mechanism. A corresponding sentence was added to the text (Pg 27, Ln 17-24).

"... The results show a significant increase in the computational cost relative to the meteorology only simulation for the same model domain. The comparison of 'transp. only' and '(full)' for Case A ( Table 2) shows that the transport of additional scalar variables is even more expensive compared to the computation of the chemical transformation. While the increase in computational costs for the transport increases linearly with the number of compounds, this is not the case for computation of chemical conversion. As described by Verwer et al. (1999) the efficiency of the applied Rosenbrock solver increases fortunately with increasing size of the chemical mechanism. Comparison of 'transp. only' of case B with 'full' suggests that for the PHSTAT mechanism the computational expense for the transport of the additional scalars is almost the same as for the computation of the chemical transformation..."

*Comment(23) Overall, the authors show horizontal variations in concentrations in Figs. 8 and 10 and vertical profiles from 0 m to 2,500 m in Figs. 3, 4, 5, and 9. However, they did not show the vertical profiles in the bottom layers (i.e., below 50 – 100 m), even though there seems very sharp vertical gradients. This might be of interest. Is it possible for the authors to show the horizontal distributions of concentrations near the top of urban canopy (at several ten meters?) to compare and discuss the differences from those at 5 m of Fig. 6, for example? Large scale models can only simulate the concentration above the urban canopy and many of the urban observation points exist on the roof of buildings. It is very informative for large scale modellers to show the difference in concentrations between the street canyon and urban canopy top.*

**Author's resp:** In figure 3 and 4, vertical profiles of meteorology, pollutant concentration and pollutant fluxes are provided from surface to 3 kilometre above ground. Sufficient explanation of these profiles is already provided in the text. However, a detailed analysis of the urban canopy processes is out of the scope of this paper.

**References**

Verwer, W G., Spee, E J., Blom, J G., Hundsdorfer, W., 1999:A second order Rosenbrock method applied the photochenmical dispersion problems.*Journal of Atmospheric Chemistry*,**37**,. doi:10.1137/S1064827597326651.
Wild, O., Zhu, X., Prather, M. J., 2000:Fast-J: Accurate Simulation of In- and Below-Cloud Photolysis in Tropospheric Chemical Models.*IAM Journal on Scientific Computing*,**20**, 1456-1480. doi:10.1023/A:1006415919030.

---

## Author Comment (AC3) · 4 Dec 2020

[11pt, a4paper]article [a4paper, left=3.5cm, right=2.5cm, top=2.0cm, bottom=2.5cm]geometry amssymb,amsmath [utf8]inputenc [T1]fontenc enumerate fancyhdr graphicx mathptmx units [authoryear]natbib hyperref pgfgantt microtype
**Response to comments of the anonymous Referee #2, "RC2"**

The paper describes the new PALM6.0 model system, an extension of the existing PALM model with on-line gas-phase chemistry (several schemes, with different levels of complexity) and deposition. The functioning of the system is evaluated for the different chemical schemes for test cases in Berlin. In addition the computational costs of the different levels of complexity are analysed. The paper is quite well-structured and complete. Case studies are interpreted in detail and compared to observations. Figures are relevant and to the point. All relevant aspects are described. However, at some places further clarification is needed. Also the English needs improvement. All in all I compliment the authors with their nice work and I'm looking forward to the final version.

**Author's resp:** First of all, we want to thank the reviewer for the effort in reviewing our paper and the valuable detailed comments, which helped us to improve the paper. Please note that page and line numbers mentioned in responses to comments correspond to the updated version of the manuscript.

**Detailed comments**
Comment: Abstract Since you name the chemical mechanisms at the end of the abstract you can also directly name them in the beginning of the abstract and rank them in complexity. Naming of emission mode is not so relevant here. The authors state that this is the first paper with complex gas phase chemistry at this high resolution in an on-line coupled model for an urban geometry. However, the feedback of modelled concentrations on meteorology is not elaborated here. The authors could say a few words on this.

Author's resp:: We have modified the abstract accordingly.

**Comment: P3: Li et al 2016 also included rather detailed chemistry, but indeed more simple than CBM-IV**

**Author's resp:**: A reference to Li et al. (2016) was already included in the submitted paper. However, we agree that the reference to this paper should be more visible and the sentence "The NCAR LES model with coupled MOZART2.2 chemistry (Kim et al., 2012) includes a quite detailed description of isoprene oxidation and its products. This model was also applied by Li et al. (2016) in order to investigate turbulence-driven segregation of isoprene over a forest area with a grid with of 150 m" has been added (Pg 3, Ln 13-14).

Comment: P7: CBM-IV is replaced by more detailed CB5 and CB6 mechanisms, so not that widely used any more, but you could give the valid argument that it is the lightest version of 'full' gas-phase chemistry.

Author's resp: We modified the text accordingly (Pg 7,Ln 15-16).

*Comment: P8 I20 confusing sentence, compounds are not all in the particulate phase* **Author's resp:** We changed the sentence to " Currently the full SALSA implementation in PALM includes the following chemical compounds in the particulate phase: sulphate ( $SO_4^{2+}$ ), organic carbon (OC), black carbon (BC), nitrate ( $NO_3^{-}$ ), ammonium GMDD
 $(NH_4^+)$ , sea salt, dust and water  $(H_2O)$ " (Pg 8, Ln 12-14).

Comment: P11 Numerical set-up description. The authors switching between description of case, used observations and numerical set-up makes it more difficult to follow or look things up.

**Author's resp:** We have re-organised this section completely. Subsection 3.1 also renamed as "Modelled episode and modelling domain". Two further subsections have been introduced in order to improve the readability of this section (Pg 11-14 Ln -).

Comment: P11 L8 It was not a weekend, therefore, emissions from the traffic were not affected by the reduced trafic Which reduced traffic? You mean that the day is a working day with the corresponding traffic activity, which is higher than in a weekend. The topic is addressed better on p 11 and p13. This sentence is confusing **Author's resp:** The sentence has been modified as "July 17 2017 was a Monday, therefore, the diurnal cycle of the traffic emissions can be described by a typical weekday with relative maxima during the morning and evening rush hours." (Pg 11, Ln 6-7).

Comment: P 12 L3-4 in one sentence 50 and 60 m used, why this criterium? What is a few?

**Author's resp:** We have modified the sentence and mentioned the number of high buildings: 'With the exception of 10 buildings with a height between 45 m and 65 m and two buildings, which are higher than 70 m, the building heights in the simulation domain are between 5 and 45 m." (Pg 11, Ln 15-16).

Comment: P13 Observed NO, NO2 and O3 from Berlin city, are these the stations in section 3.2? Did you interpolate or take one value for the domain?
**Author's resp:** Yes, we used values from this network. This is explained more clearly now in the text and we added a remark, that all grid points of the domain are initialised with identical pollutant profiles (Pg 14, Ln 17-18).

*Comment:* P15: This part about chemistry needs some reformulation and better explanations. R3 should also contain an M on the left side. **Author's resp:** Thanks for making us aware of the missing M. The reaction R3 and some text in this section has been updated (Pg 16, Ln 9).

Comment: P16 I6 The authors state that NO and NO2 do not lead to a net gain in O3. But that is not completely true, the final photostationary equilibrium depends on NO2/NO ratio and changing emissions/concentration leads to a change in O3, as correctly stated on p18/12. VOC plays an additional role. For an urban area with NOx abundance, VOC is mostly the limiting factor for O3 formation. Also in R4, the meaning of RH and RO2 are not explained in the text. The additional impact of VOC is indeed visible in Figure 10.

**Author's resp:** We have added a remark that ozone production is possible by emission and subsequent photolysis of primary  $NO_2$ . RH and RO2 are now explained (Pg 17, Ln 6-13).

Comment: P17, section 4.3 Spatial distribution of pollutants: why do you switch to CBM4 here, whereas SMOG was the chosen to be the default chemical scheme? 4.2 was with SMOG (as I understood from the context and the model description: SMOG was chosen. . .p13 l8)

**Author's resp:** We are sorry that we did not mention properly that the baseline case was performed with CBM4. We have added the sentence "Assuming that CBM4 is more accurate than the more simple mechanisms due to its more complete representation of atmospheric chemistry, the baseline simulation of this study was
performed with CBM4." Furthermore, the mechanism used is now also mentioned in the figure captions (Pg 13, Ln 27-29).

Comment: P22 I 28 Downwelling in the entrainment zone: I would call this entrainment, mixing in of air when the boundary layer rises. See also p24 last sentence. **Author's resp:** This sentence has been rephrased (Pg 23, Ln 13-14).

Comment: P24 I 18 Missing emission sources: would you expect that the contributions from industry, household and BVOC would have a significant peak in the evening in summer? I doubt this.

Author's resp: We agree and have removed this sentence (Pg 25, Ln 19-20).

*Comment: P25 Section 4.6 Numerical efficiency test, are case A and B related to a specific location? Is domain A included in B, for which day?* **Author's resp:** Yes, the smaller model domain of Case A is included in B. This is now also mentioned in the text (Pg 27, Ln 8-9).

Comment: P27 Concluding remarks. Now that the chemistry schemes are compared in terms of performance with respect to observations and computational effort, could you conclude whether SMOG or CBM-IV would be the best default option? Especially in the light of the further complexity of the model (BVOC, SALSA) and the last sentence. How does your conclusion relate to practices in other LES models? **Author's resp:** We added a further paragraph to the conclusions, which addresses this issue (Pg 29, Ln 16-24).

Comment: Typo's/text corrections a. P10 l6 sentence with mostly sunny with. . . GMDD
b. L 16 Ceilometere

c. P22 I 31-32 Although. . .., however. . . Use one of the two d. P14 output-> put out/written to file. Output was exported to file every 10 minutes. .

**Author's resp:** Thanks for making us aware of typos and suggestions for improvement. The text is updated (a. Pg 11, Ln 5; b. Pg 12, Ln 11; c. Pg 23, Ln 17; d. Pg 15, Ln 4).

Comment: P10: Meteo from web sources: this is meteo from the airport published at a website I assume. When I read the sentence I'm in doubt of the data source and quality.

**Author's resp:** We have replaced the reference by a better reference, which refers to the same data. We have also mentioned that the metéo data are from Berlin Tegel airport (Pg 12, Ln 8-9).

*Comment: P11, p12 Some brackets missing in references* **Author's resp:** Missing brackets added. Pg 12, Ln 9,10

Comment: P12 L9-11 confusing and..and.. with respectively, unclear what is meant exactly

**Author's resp:** There was indeed something wrong with this sentence. We replaced it by "Trees resolved by the canopy model are characterised by the three-dimensional leaf area density per unit volume (LAD). For the model configuration used here, LAD is considered for, i.e. up to a maximum height of 40 m above the ground and assumes values up to  $3.1 \text{ m}^2 \text{ m}^{-3}$  with an average value of  $0.44 \text{ m}^2 \text{ m}^{-3}$ ." (Pg 12, Ln 1-4).
**References**

- Kim, S.-W., Barth, M C., Trainer M., 2012:Influence of fair-weather cumulus clouds on isoprene chemistry. *Journal of Geophysical Research: Atmospheres*, **117**. doi:10.1029/2011JD017099.
- Li, Y., Barth, M.C., Chen, G., Patton, E.G., Kim, Si-Wan., Wisthaler, A., Mikoviny, T., Fried, A., Clark, R., Steiner, A L., 2016:Large-eddy simulation of biogenic VOC chemistry during the DISCOVER-AQ 2011 campaign. *Journal of Geophysical Research: Atmospheres*, **121**(13,) 8083-8105. doi:10.1002/2016JD024942.

---

## Author Response (AR1)

Dated: 29 December 2020

Basit Khan
Karlsruhe Institute of Technology (KIT)
Garmisch-Partenkirchen, 82467
Germany

Dr. Astrid Kerkweg

Editor

Geoscientific Model Development

**Subject: Revision of manuscript # gmd-2020-286**

Dear Dr. Kerkweg,

Please find below point-by-point reply to the comments of the two referees. In addition to comments, we have made several changes to improve the readability of the manuscript. All changes are visible in the marked-up version (copied below) of the manuscript.

Thanks again for your time. Please let us know if you need further information.

Best wishes,

Basit Khan and Co-authors.

Karlsruher Institut für Technologie (KIT)
Institut für Meteorologie und Klimaforschung
Atmosphärische Umweltforschung (IMK-IFU)

 [Transport Processes in the Atmospheric Boundary Layer]
KIT Campus Alpin, Kreuzeckbahnstr. 19,
82467 Garmisch-Partenkirchen,Germany.
Email : basit.khan@kit.edu
Phone : +49 (0)8821 183 323
Mobile: +49 (0)1523 7244 535

**Response to comments of anonymous Referee #1, "RC1" for the manuscript entitled "Development of an atmospheric chemistry model coupled to the PALM model system 6.0: Implementation and first applications"**

**General Comments:**

*Comment (1): The current manuscript describes a new city-scale LES-Chem model, PALM-4U and presents test simulation results with a variety of chemistry modules. It is indeed important to have a variety of chemistry modules from simple to complex in one model, so that each user can allocate limited computational resources to his/her own specific targets. Some might need higher resolutions or longer time integrations, while some might need very accurate chemical modules for coarser resolutions or shorter time integrations. The quality of their work meets the standard of GMDD but there are several issues remained to improve the presentation of manuscript as listed in the following.*

**Author's resp:** We would like to thank the Referee for the effort in reviewing our manuscript and the valuable detailed comments, which helped us to improve the paper. Please note that page and line numbers mentioned in author's responses/author's changes correspond to the updated version of the manuscript.

**Specific Comments:**

*Comment (1) The country names are missing in the affiliations of co-authors from #2 to #6.*
**Author's resp:** We agree that country names should be added with affiliations.
**Author's changes:** Country name has been added from affiliation #2 to #6.

*Comment(2) There are several abbreviations in abstract without being defined, such as PALM, PARAMETERIZED, CBM4, SMOG, and PHSTAT. PALM is their model name but it is never spelled out throughout the manuscript.*
**Author's resp:** PALM, and chemistry mechanisms have now been defined when the first time they appear in the text. Emission mode 'PARAMETERIZED' is defined briefly in the abstract and then in greater detail in section 2.3.
**Author's changes:** a) "...PALM model system 6.0 (formerly an abbreviation for Parallelized Large-eddy Simulation Model and now an independent name)... ", (Pg 1, Ln 2)
b) "...from the photostationary state (PHSTAT) to mechanisms with a strongly simplified VOC chemistry(e.g. the SMOG mechanism from KPP) and the Carbon bond mechanism (CBM4, Gery et al. (1989)), which includes major pollutants namely $O_3$, NO, $NO_2$, CO, a more comprehensive, but still simplified VOC chemistry and several products ... ",(Pg 1 Ln 8-10).

*Comment(3) There are more abbreviations in the entire manuscript, such as MITRAS, ASMUS, SALSA, etc. Better to spell them out when they appeared first time, or make a table of nomenclature.*
**Author's resp:** Abbreviations have now been spelled out in the text.
**Author's changes:** "...and trees e.g. MITRAS (the microscale obstacle resolving transport and stream model), (Salim et al., 2018) and ASMUS (A numerical model for simulations of wind and pollutant dispersion around individual buildings), (Gross, 1997),...",(Pg 2, Ln 31-32)
b) "... SALSA (A sectional aerosol module for large scale applications), (Kokkola et al., 2008)

implemented in PALM ...", (Pg 4, Ln 9)

*Comment(4) The relationship or difference between PALM and PALM-4U is unclear in the entire manuscript. Sometimes the author write PALM, but sometimes PALM-4U. It can be read that PALM consists of PALM-4U and the chemistry module was in PALM-4U, not PALM (Sect.2.1 and Fig. 1). However, in Sect. 2.5.5, it seems that the chemistry module was with PALM, and PALM-4U was not appeared. In Sect. 2.5.6 as well. It is a bit confusing. Please clarify the relationship or difference between the two models and be accurate in the definition throughout the manuscript.*

**Author's resp:** PALM is primarily an LES code with an optional RANS mode. The PALM model comprised of a dynamic core and various embedded models. PALM-4U is the conceptual framework of the model components embedded/coupled to PALM which are specifically designed and/or used for urban applications. The description of PALM and PALM-4U has been rephrased in the text (abstract and in section 2.1) so that to make the difference between the two more clear and easy to understand. Subsections 2.5.5 and 2.5.6 do not exist in the manuscript, therefore, the issue raised by the reviewer cannot be addressed.

**Author's changes:** "... The new chemistry model is implemented in the PALM model as part of the PALM-4U components (PALM for urban applications) which are designed for application of PALM model in the urban environment (Maronga et al., 2020)...", (Pg 1, Ln 4-5); "The PALM model system 6.0, consists of the PALM core and PALM-4U (PALM for urban applications) components (Maronga et al., 2020) that have been added to PALM under the MOSAIK (Model-based city planning and application in climate change) project (Maronga et al., 2019), one of which is the chemistry module described in this paper.", (Pg 4, Ln 20-22); "Schematic representation of the chemistry model (PALM-4U component) of the PALM model system. The arrows show interaction between PALM model core, the chemistry driver module and sub-modules. The dashed box indicates the chemical preprocessor which generates subroutines to solve chemical reactions.",(Pg 5, Figure 1 caption)

*Comment(5) P.4 Ln. 16: "Optical cloud and rain water" may require explanation. It was written later that cloud microphysics was not implemented, so what are they?*

**Author's resp:** We never wrote that cloud microphysics was not implemented. Since PALM version 4.0 a bulk liquid-phase two-moment microphysics scheme is available which predicts rain droplet number concentration and rain water mixing ratio. To avoid any confusion, the text has been slightly rephrased.

**Author's changes:** " ... the three velocity components ($u$, $v$, $w$) on staggered Arakawa C grid, and four scalar variables namely potential temperature ($\theta$) , water vapour mixing ratio ($qv$), a passive scalar s and the subgrid-scale turbulent kinetic energy (SGS-TKE) $e$ (in LES mode) (Maronga et al., 2019, 2020)...", (Pg 4, Ln 24-26).

*Comment(6) Figure 1: Arrows are ambiguous. What's the difference between grey arrow and green arrows? What's the difference between the green arrow with single direction and bi-direction? It seems that emission to chemistry driver is one-way, but there is a direction from chemistry toward emission. What process is it?*

**Author's resp:** Green and thick grey arrows were added to improve presentation of the flow chart. However, to avoid any confusion, Figure 1 has been further improvised.

**Author's changes:** All arrows are now drawn in grey colour. The thin arrows show the general flow and direction of the data within the Chemistry model and its components whereas thick grey arrow indicates coupling of the Chemistry driver with the PALM model core (Pg 5).

*Comment(7) P.6, L.15 "deposition, scavenging" -> "deposition and scavenging"*

**Author's resp:** The text is updated.

**Author's changes:** ".... (i.e. deposition and scavenging).", (Pg 6, Ln 15).

*Comment(8) P.8, L.13 "Fast-J" may need reference. If it is an abbreviation, please spell it out.*
**Author's resp:** The text is updated and an appropriate reference has been added.
**Author's changes:** "... More extensive photolysis schemes such as the Fast-J photolysis scheme (Wild et al. , 2000) that are based on the radiative transfer modelling will be included in the future...", (Pg 8, Ln 5-6).

*Comment(9) P.8, L.20-21; for aerosol phase, better to write sulfate (SO42-), nitrate (NO3-), and ammonium (NH4+).*
**Author's resp:** The text has been updated.
**Author's changes:** "... following chemical compounds in the particulate phase: sulphate ($SO_4^{2+}$), organic carbon (OC), black carbon (BC), nitrate ($NO_3^-$), ammonium ($NH_4^+$), sea salt, dust and water ($H_2O$) ...", (Pg 8, Ln 13-14)

*Comment(10) P.9, L.10: "following (Simpson et al., 2003)." -> "following Simpson et al. (2013)."*
**Author's resp:** The reference has been updated.
**Author's changes:** " ... calculated following Simpson et al. (2003)... ", (Pg 9, Ln 11).

*Comment(11) P.9, L.29: "The chemistry model of PALM" -> "The chemistry model of PALM-4U", right?*
**Author's resp:** Yes, this is right. the text has been updated.
**Author's changes:** "... The chemistry model of PALM-4U includes a module for reading ...", (Pg 9, Ln 30).

*Comment(12) P. 10, L.12 "modes": "sectors" are more frequently used. Please consider to rephrase.*
**Author's resp:** Thanks to point this out. We agree this sentence was slightly misleading. We have rephrased and improved the text.
**Author's changes:** "... More detailed traffic emission data can be provided in gridded form in PALM-specific NetCDF files (Maronga et al., 2020). LOD 1 emissions are gridded annual emission data for each sector (e.g. industry, domestic heating, traffic), which will be temporally disaggregated using sector-specific standard time factors...", (Pg 10, Ln 11-13 ).

*Comment(13) Sect. 3.1 "numerical set-up" includes several sentences which should be described in different sections.*
*a) P. 11, L. 2: "Details of the dynamics core ... Maronga et al. (2020)" better to be moved to a model description section, Sect. 2.x.*
*b) 2nd paragraph of Sect. 3.1, the first two sentences "Observations from ... a 24 hour run." and "The ceilometer observations ... the diurnal cycle" and the latter two sentences "Fig. 2 shows ..." and "The horizontal grid spacing..." are not relating with each other. The description of weather by Ceilometer observation was already mentioned in the 1st paragraph of Sect. 3.1. Better to reorganise the 1st and 2nd paragraph of Sect. 3.1.*
*c) P. 11, L. 26, "A third order Runge-Kutta, ... (Wicker and Skamarock, 2002)" are already written previously.*
**Author's resp:** We have re-organised section 3 completely.
**Author's changes:** a) "Details of the dynamic core . . . ..", has been moved to the model description section 2.1 (Pg 4, Ln 32-33).
b) Subsection 3.1 renamed as "Modelled episode and modelling domain", subsection 3.2 is renamed as "Observational data" and a third subsection "3.3 Model configuration and initialisation" has been introduced in order to improve the readability of this section" (Pg 11-14 Ln – ).
c) We would like to retain this. The first time Runge-Kutta and Wicker and Skamarock appeared in the general model description while, second time, both appeared specifically as part of the model

setup for simulations.

*Comment(14) P. 11, L.12: Spell "TU" out here.*
**Author's resp:** The abbreviation (TU) is removed and the actual name "Technical University" has been used instead
. **Author's changes:** "... Charlottenburg building of the Technical University of Berlin ...", (Pg 12, Ln 15).

*Comment(15) P.11, L.165: "Ceilometere" -> "Ceilometer"*
**Author's resp:** The spelling has been corrected (Pg 12, Ln 13).

*Comment(16) P. 12, L. 2: Probably "(Resler et al., 2017; Maronga et al. 2020)" looks better.*
**Author's resp:** Both of the citations have been updated in the text.
**Author's changes:** "... the urban-surface model (Resler et al., 2017; Maronga et al., 2020).", (Pg 13, Ln 18).

*Comment(17) P. 12, L.16: "Monin-Obukhov Similarity Theory (MOST)" mentioned already several times in the previous locations. Define MOST when it is first appeared and use MOST in the following locations.*
**Author's resp:** The Monin-obukhov similarity theory is abbreviated as MOST, the first time it appeared in the text, and then only "MOST" used in the following text.
**Author's changes:** "Monin-Obukhov similarity (MOST) is assumed between every individual surface ....", (Pg 4, Ln 28).

*Comment(18) P.13, L.25-35: How the authors set the boundary conditions for chemical species are not explained.*
**Author's resp:** The description of the chemical boundary conditions on lateral, top and bottom boundaries has been further improved to make it more clear and easy to understand.
**Author's changes:** a) "At the bottom boundary, a Dirichlet condition is applied to flow, $\theta$, and $q$ whereas a Neumann condition is applied to $e$, $p$ and chemical compounds.",(Pg 14 Ln 29-30).
b) "At the top boundary, Dirichlet boundary conditions are applied to flow and and $p$ only, initial gradient is applied to $\theta$ while Neumann boundary conditions are applied to $q$ and chemical compounds.", (Pg 14, Ln 31-32).

*Comment(19) Sect. 3.2: Please provide the heights of the observation sites, Wedding and Hardenbergplatz. In urban locations, the observation points could be on the roof of building. If this is the case, comparison between the simulated 5-m height concentration and the observed roof-top concentration are inconsistent.*
**Author's resp:** The average height of air quality sensors is around 4 m above ground. Therefore, comparison between observations and model data which is extracted 5 m above ground is consistent. The text has been updated in section 3.2.
**Author's changes:** "... average height of the air quality sensors at both stations is 4 m above ground...",(Pg 12, Ln 28).

*Comment(20) Sect. 4.2: The chemistry module and the grid point used to depict Fig. 4 was missing. Probably CBM4 and Hardenbergplatz, though.*
**Author's resp::** These are the spatial mean profiles of concentration and fluxes. The text and figure caption has been updated by adding "mean" and "simulated with CBM4 mechanism".
**Author's changes:** "Figure 3. Vertical profiles of a) potential temperature, b) mixing ratio, c) wind speed and d) wind direction, at different times from morning to midnight on 17 July 2017. The horizontal bars in a) indicate the boundary-layer height derived from ceilometer observations.", ( Pg 15, Ln–).

"... Figure 4 shows mean profiles of concentrations and vertical fluxes of NO, NO2, O3and CO for the selected times of the diurnal cycle on 17 July 2017, simulated with the CBM4 mechanism...", (Pg 15, Ln 17-18).

*Comment(21) P. 24, L. 5: "Figure 10" -> "Fig. 10"*
**Author's resp:** The inconsistency in the use of '*Figure′*', and '*Fig.′*', has been removed. In the beginning of the sentence we use '*Figure′*', whereas within the sentence we use '*Fig.′*'. The use of both is now consistent throughout the manuscript.

*Comment(22) Table 2: It is quite reasonable that the CPU time of transport only without meteorology of CBM4 was 10 times that of PHSTAT (310/30) because the number of tracers is also 10 times (32/3). However, with chemical reactions, why the CPU time of CBM4 was still 10 times that of PHSTAT (550/50) even though the number of chemical reactions of CBM4 (81) is 40 times that of PHSTAT (2). Is it because only two reactions with KPP requires as much CPU time as 81 reactions?*
**Author's resp:** It is correct that CBM4 is 10 times more expensive than PHSTAT both for 'transport-only' and for 'Full' (transport + reactions) simulations. The relative increase of the computational expense for the solution of the chemistry equations does indeed decrease with increasing number of chemical compounds. As explained in section 3.4 of Verwer et al. (1999), the efficiency of the applied Rosenbrock solver increases with increasing size of the chemical mechanism. The paragraph has been re-written with some additional text to make analysis more clear and easy to understand.
**Author's changes:** "The results show a significant increase in the computational cost relative to the meteorology only simulation for the same model domain. The comparison of 'transp. only' and '(full)' for Case A ( Table 2) shows that the transport of additional scalar variables is even more expensive compared to the computation of the chemical transformation. While the increase in computational costs for the transport increases linearly with the number of compounds, this is not the case for computation of chemical conversion. As described by Verwer et al. (1999) the efficiency of the applied Rosenbrock solver increases with increasing size of the chemical mechanism. Comparison of 'transp. only' of case B with 'full' suggests that for the PHSTAT mechanism the computational expense for the transport of the additional scalars is almost the same as for the computation of the chemical transformation...", (Pg 27, Ln 17-24).

*Comment(23) Overall, the authors show horizontal variations in concentrations in Figs. 8 and 10 and vertical profiles from 0 m to 2,500 m in Figs. 3, 4, 5, and 9. However, they did not show the vertical profiles in the bottom layers (i.e., below 50 – 100 m), even though there seems very sharp vertical gradients. This might be of interest. Is it possible for the authors to show the horizontal distributions of concentrations near the top of urban canopy (at several ten meters?) to compare and discuss the differences from those at 5 m of Fig. 6, for example? Large scale models can only simulate the concentration above the urban canopy and many of the urban observation points exist on the roof of buildings. It is very informative for large scale modellers to show the difference in concentrations between the street canyon and urban canopy top.*
**Author's resp:** In figure 3 and 4, vertical profiles of meteorology, pollutant concentration and pollutant fluxes are provided from surface to 3 kilometre above ground. Sufficient explanation of these profiles is already provided in the text. However, a detailed analysis of the urban canopy processes is out of the scope of this paper.

**Response to comments of anonymous Referee #2, "RC2" for the manuscript entitled "Development of an atmospheric chemistry model coupled to the PALM model system 6.0: Implementation and first applications"**

*The paper describes the new PALM6.0 model system, an extension of the existing PALM model with on-line gas-phase chemistry (several schemes, with different levels of complexity) and deposition. The functioning of the system is evaluated for the different chemical schemes for test cases in Berlin. In addition the computational costs of the different levels of complexity are analysed. The paper is quite well-structured and complete. Case studies are interpreted in detail and compared to observations. Figures are relevant and to the point. All relevant aspects are described. However, at some places further clarification is needed. Also the English needs improvement. All in all I compliment the authors with their nice work and I'm looking forward to the final version.*

**Author's resp:** First of all, we want to thank the reviewer for the effort in reviewing our manuscript and the valuable detailed comments, which helped us to improve the manuscript. Please note that page and line numbers mentioned in the author's responses/author's changes correspond to the updated version of the manuscript.

**Specific Comments**

*Comment: Abstract Since you name the chemical mechanisms at the end of the abstract you can also directly name them in the beginning of the abstract and rank them in complexity. Naming of emission mode is not so relevant here. The authors state that this is the first paper with complex gas phase chemistry at this high resolution in an on-line coupled model for an urban geometry. However, the feedback of modelled concentrations on meteorology is not elaborated here. The authors could say a few words on this.*

**Author's resp:** We received similar comments from Referee #1 also and therefore, we modified the abstract considering comments from both Referees.

**Author's changes:** a) The chemical mechanism names are spelled out the first time they appeared in the abstract; b) We considered to retain the emission mode name; c) The feedback of the chemistry on the meteorology has not yet been implemented. A sentence "Even though the feedback of model's aerosol concentrations on meteorology is not yet considered in the current version of the model, the results show the ...." has been added (Pg 1, Ln 18).

*Comment: P3: Li et al 2016 also included rather detailed chemistry, but indeed more simple than CBM-IV*

**Author's resp::** A reference to Li et al. (2016) was already included in the submitted manuscript. However, we agree that the reference to this paper should be more visible.

**Author's changes:** "The NCAR LES model with coupled MOZART2.2 chemistry (Kim et al., 2012) includes a quite detailed description of isoprene oxidation and its products. This model was also applied by Li et al. (2016) in order to investigate turbulence-driven segregation of isoprene over a forest area with a grid with of 150 m" has been added (Pg 3, Ln 13-15).

*Comment: P7: CBM-IV is replaced by more detailed CB5 and CB6 mechanisms, so not that widely used any more, but you could give the valid argument that it is the lightest version of 'full' gas-phase chemistry.*

**Author's resp:** We modified the text accordingly.

**Author's changes:** "As a representative of a 'full' gas phase mechanism the well-known Carbon-Bond-IV (CBM4) (Gery et al., 1989) is included. Although CBM4 has been replaced by the more detailed CB5 and CB6 mechanisms in the meantime, it is still applied in some models. CBM4 was implemented in PALM, since – with 32 compounds – it is the smallest of the full mechanisms.", (Pg 7,Ln 14-17).

*Comment: P8 l20 confusing sentence, compounds are not all in the particulate phase*

**Author's resp:** We agree with the comment and have changed the sentence to make it more clear and easy to understand.

**Author's changes:** "Currently the full SALSA implementation in PALM includes the following chemical compounds in the particulate phase: sulphate ($SO_4^{2+}$), organic carbon (OC), black carbon (BC), nitrate ($NO_3^-$), ammonium ($NH_4^+$), sea salt, dust and water ($H_2O$)" (Pg 8, Ln 12-14).

*Comment: P11 Numerical set-up description. The authors switching between description of case, used observations and numerical set-up makes it more difficult to follow or look things up.*

**Author's resp:** We have re-organised this section completely.

**Author's changes:** Subsection 3.1 also renamed as "Modelled episode and modelling domain", subsection 3.2 is renamed as "Observational data" and a third subsection "3.3 Model configuration and initialisation" has been introduced in order to improve the readability of this section" (Pg 11-14 Ln – ).

*Comment: P11 L8 It was not a weekend, therefore, emissions from the traffic were not affected by the reduced traffic Which reduced traffic? You mean that the day is a working day with the corresponding traffic activity, which is higher than in a weekend. The topic is addressed better on p 11 and p13. This sentence is confusing*

**Author's resp:** We agree with the comment. The sentence has been modified.

**Author's changes:** "July 17 2017 was a Monday, therefore, the diurnal cycle of the traffic emissions can be described by a typical weekday with relative maxima during the morning and evening rush hours." (Pg 11, Ln 6-7).

*Comment: P 12 L3-4 in one sentence 50 and 60 m used, why this criterium? What is a few?*

**Author's resp:** We have modified the sentence and mentioned the number of high buildings.

**Author's changes:** "With the exception of 10 buildings with a height between 45 m and 65 m and two buildings, which are higher than 70 m, the building heights in the simulation domain are between 5 and 45 m." (Pg 11, Ln 16; Pg 12, Ln 1-2).

*Comment: P13 Observed NO, NO2 and O3 from Berlin city, are these the stations in section 3.2? Did you interpolate or take one value for the domain?*

**Author's resp:** Yes, we used values from this network. This is explained more clearly now in the text and we added a remark, that all grid points of the domain are initialised with identical pollutant profiles.

**Author's changes:** "The initial profiles of pollutant concentration are based on the mean observed near surface concentrations of NO, $NO_2$ and $O_3$ from the stations of the BLUME network (section 3.2). Initial concentrations of NO, $NO_2$ and $O_3$ above 495 m were set to 0.0, 2.0 and 40.0 ppb respectively. Considering the strong impact of traffic emissions on local pollutant concentrations all grid points of the model domain were initialised with identical pollutant profiles.", (Pg 14, Ln 16-19).

*Comment: P15: This part about chemistry needs some reformulation and better explanations. R3 should also contain an M on the left side.*

**Author's resp:** Thanks for making us aware of the missing M. The reaction R3 and some text in this section has been updated.

**Author's changes:** "$O_2 + O(^3P) + M \rightarrow O_3 + M$  (R3) .", (Pg 16, Ln 9 ).

*Comment: P16 l6 The authors state that NO and NO2 do not lead to a net gain in O3. But that is not completely true, the final photostationary equilibrium depends on NO2/NO ratio and changing emissions/concentration leads to a change in O3, as correctly stated on p18l12. VOC plays an additional role. For an urban area with NOx abundance, VOC is mostly the limiting factor for O3 formation. Also in R4, the meaning of RH and RO2 are not explained in the text. The additional impact of VOC is indeed visible in Figure 10.*

**Author's resp:** We have added a remark that ozone production is possible by emission and subsequent photolysis of primary $NO_2$. RH and RO2 are now explained.

**Author's changes:** a) " ..unless additional $NO_2$ is supplied (e.g. primary $NO_2$ from traffic emissions) and $O_3$ is formed by reaction R2 and R3."(Pg 17, Ln 6-7 ); b) "where RH stands for any explicitly described or lumped non-methane hydrocarbon and RO2 represents any organic peroxy radical.", (Pg 17, Ln 12-13 )

*Comment: P17, section 4.3 Spatial distribution of pollutants: why do you switch to CBM4 here, whereas SMOG was the chosen to be the default chemical scheme? 4.2 was with SMOG (as I understood from the context and the model description: SMOG was chosen. . .p13 l8)*

**Author's resp:** We are sorry that we did not mention properly that the baseline case was performed with CBM4. We have now added a sentence to make it clear and easy to understand. Furthermore, the mechanism used is now also mentioned in the figure captions

**Author's changes:** "Assuming that CBM4 is more accurate than the more simple mechanisms due to its more complete representation of atmospheric chemistry, the baseline simulation of this study was performed with CBM4.", (Pg 13, Ln 28-29).

*Comment: P22 l 28 Downwelling in the entrainment zone: I would call this entrainment, mixing in of air when the boundary layer rises. See also p24 last sentence.*

**Author's resp:** This sentence has been rephrased.

**Author's changes:** "During the daytime specially and in particular, in the morning hours, entrainment of $O_3$ from the residual layer during the growth of the boundary layer contributes to ...", (Pg 23, Ln 13-14).

*Comment: P24 l 18 Missing emission sources: would you expect that the contributions from industry, household and BVOC would have a significant peak in the evening in summer? I doubt this.*

**Author's resp:** We agree and have removed this sentence. A new sentence has been added to explain the under-representation of the chemical species.

**Author's changes:** "We utilized only parameterized traffic emissions, which may under-represent the emissions at the considered locations in the evening hours.", (Pg 25, Ln 20-21).

*Comment: P25 Section 4.6 Numerical efficiency test, are case A and B related to a specific location? Is domain A included in B, for which day?*

**Author's resp:** Yes, the smaller model domain of Case A is included in B. This is now also mentioned in the text.

**Author's changes:** "The model domain for the Case A simulations is smaller part of the model domain shown in Figure 2 with the centre located at the Ernst-Reuter-Platz.", (Pg 27, Ln 8-10).

*Comment: P27 Concluding remarks. Now that the chemistry schemes are compared in terms of*

*performance with respect to observations and computational effort, could you conclude whether SMOG or CBM-IV would be the best default option? Especially in the light of the further complexity of the model (BVOC, SALSA) and the last sentence. How does your conclusion relate to practices in other LES models?*

**Author's resp:** We added a further paragraph to the conclusions, which addresses this issue.

**Author's changes:** "Although the maximum ozone concentration and the time of its occurrence is somewhat better reproduced with CBM4 than with the SMOG mechanism it is difficult to give a final recommendation for an optimum mechanism. For some applications – in particular for low VOC conditions – SMOG or even the photostationary equilibrium may be sufficient while for other simulations the application CMB4 is simply not possible due to its computational demand. However, when using strongly condensed mechanisms like SMOG with only one single VOC compound the user must always keep in mind that the simulated concentrations of ozone and further oxidation products depend strongly on the rate constant for the reaction of this single lumped VOC with OH. As already elaborated by Middleton et al. (1990) the reactivity of the lumped VOC depends on the VOC mix and therefore on the local emissions. On the other hand CMB4 is still too simple for investigations with special focus on VOC chemistry or a prediction of semi- and non-volatile organics, which are required for the SALSA aerosol model. In this case more detailed or advanced chemistry models than CBM4 may be required which can be easily added since the flexible configuration of the chemistry in PALM allows the easy implementation of new mechanisms.", ( Pg 29, Ln 5-15).

*Comment: Typo's/text corrections*
*a. P10 l6 sentence with mostly sunny with. . .*
*b. L 16 Ceilometere*
*c. P22 l 31-32 Although. . .., however. . . Use one of the two*
*d. P14 output-> put out/written to file. Output was exported to file every 10 minutes. . ..*

**Author's resp:** Thanks for making us aware of typos and suggestions for improvement. The text is updated.

**Author's changes:** (a. Pg 11, Ln 4; b. Pg 12, Ln 13; c. Pg 23, Ln 17; d. Pg 15, Ln 4).

*Comment: P10: Meteo from web sources: this is meteo from the airport published at a website I assume. When I read the sentence I'm in doubt of the data source and quality.*

**Author's resp:** We have replaced the reference by a better reference, which refers to the same data. We have also mentioned that the metéo data are from Berlin Tegel airport.

**Author's changes:** "In addition to routine observations of near surface temperature, cloud cover, and wind speed and direction at the Berlin Tegel airport from the open access Climate Data Center (CDC) of the German Weather Service (Deutscher Wetterdienst, 2020, last accessed on 30.07.2020), we also analysed radiosonde data from Lindenberg (Oolman, 2017, last accessed on 30.07.2020), ....", (Pg 12, Ln 10-12).

*Comment: P11, p12 Some brackets missing in references*

**Author's resp:** None of the brackets were missing in the references on page 11 and 12. There was only one additional bracket in one of the reference that has been removed.

**Author's changes:** "(Scherer et al., 2019a; Wiegner et al., 2002)", (Pg 12, Ln 17)

*Comment: P12 L9-11 confusing and..and.. with respectively, unclear what is meant exactly*

**Author's resp:** There was indeed something wrong with this sentence. We have modified the text to make it clear and easy to understand.

[revised manuscript text omitted]
 (**?**) last accessed on 30.06.2020. If a surface element has vegetation, pavement or water, it is classified as a natural-type surface and it is treated by the land-surface model, while building surfaces are treated by the urban-surface model Resler et al. (2017); Maronga et al. (2020). With the exception of a few higher buildings above 50 m, most of the building height in the simulation domain is between 5 to 45 m whereas only few exceeds 60 m.

The street types shown in Figure 2b are based on OpenStreetMap (OpenStreetMap contributors, 2017). The model domain

10  includes 13 types of narrow, one-way and two-way roads. The width of the road/street is relevant as for the emission mode with LOD=0 (parameterized mode) emission factors vary with the street type. For the main streets we used an emission factor 1.667, while for the side streets the emission factor was 0.334.

The simulation domain contains five types of vegetation categories that include grass and shrubs of different height. The three-dimensional vegetation parameters leaf-area density (LAD) representing leaves and branches and trunk have four vertical

15  levels ranging from 0 to 3 m$^2$ m$^{-3}$ and 0 to 80 m$^2$ m$^{-3}$, respectively. 
[revised manuscript text omitted]